# Synovial matrix turnover controls immune cell spatial patterning in inflammation resolution

Jean-Baptiste Richard [1], Anna Hoyle [1], Molly Bower[1], Shihong Wu [1], Leia Worthington [1], Sarah Davidson[1], Zofia Varyova[1], Caroline Morrell [1], Mathilde Pohin[1], Barbora Schonfeldova[1], Zhi Yi Wong[1], Lucy MacDonald[2], Mariola Kurowska-Stolarska [2], Stephanie G Dakin [3], Irina Udalova[1], Calliope A Dendrou[1], Anja Schwenzer [1], Christopher D Buckley[1] & Kim S Midwood [1]✉

## Abstract

**Immune-mediated inflammatory diseases remain plagued by poor treatment responses and lack curative therapies. Convergent findings suggest a role for the stromal compartment and extra-cellular matrix composition dysregulation. Using rheumatoid arthritis as a model, we define an analytical pipeline combining transcriptomic, proteomic and degradomic analysis to characterise disease activity-specific matrix perturbations. This revealed synergistic contributions from fibroblasts and myeloid cells to matrix composition, with fibroblast subsets defining distinct sub-synovial niches through distinct matrix expression profiles. Transcriptional dysregulation of collagen VI was found to be a feature of RA activity, with collagen VI protein accumulation linked to remission-associated states. Spatial analysis and in vitro migration showed collagen VI inhibits immune ingress, confining infiltrating cells to perivascular pockets termed "COL6 dark" zones. Matrix degradation-associated monocytes were found at the leading edge of these zones, expanding immune-permissive niches, and releasing RA-associated collagen VI fragments. Our work reveals how dynamic matrix remodelling can in turn limit, and enable, cell immigration in RA, identifying a new mechanism controlling tissue-level disease activity.**

**Keywords** Multi-omics; Immune-mediated Diseases; Extracellular Matrix; Rheumatoid Arthritis; Immune Resolution
**Subject Category** Immunology

## Introduction

The incidence of Immune-Mediated Inflammatory Diseases (IMIDs) has sharply increased over the last three decades (Wu et al, 2023), requiring the development and implementation of novel preventative strategies to curb this trend. Because of their poorly understood complexity and multifactorial nature (García et al, 2020), IMIDs are often plagued by poor response rates to available treatments and a complete lack of curative therapeutic avenues (Melville et al, 2020; Raine et al, 2021). Their rising incidence adds both urgency to a significant unmet clinical need, and pressure to healthcare resources already stretched to their limit in the wake of the COVID-19 pandemic (Burau et al, 2024).

While the primary research focus surrounding IMIDs has been on immune cells, therapeutic strategies targeting immune cells and their products have been unsuccessful in providing sustained remission across all patient subsets, and fail to promote curative amelioration of the pathology, notably at the level of tissue repair (McInnes and Gravallese, 2021). Recent observations have described clear roles for dysregulation of the stromal compartment in the affected tissues. Here again, the focus has been placed on pseudo-immune roles for these cells through their uncontrolled secretion of pro-inflammatory cytokines and chemokines, creating an inflammatory milieu which sustains the chronicity of IMIDs (Zhou et al, 2024). Little has been done however to define dysregulations in one of the primary functions of the stromal compartment in IMIDs, the establishment and maintenance of extracellular matrix (ECM) networks, through which are defined broad tissue-level architectures and sub-tissue-level micro-architectures.

Matrix composition dysregulation is at the core of many, if not all, human conditions. This stretches from canonical musculoskeletal (MSK) conditions such as Osteogenesis Imperfecta (Marini et al, 2017), to fibrosis, where it is a core defining feature (Henderson et al, 2020), or the modulation of tumour growth and treatment responses in certain types of cancer (Henke et al, 2020; Yuan et al, 2023). While the onus has typically been placed on MSK and fibrotic conditions for the pathogenic consequences of matrix dysregulation, there is increasing evidence for its importance in IMIDs, for example, in Inflammatory Bowel Diseases (IBD) (Shimshoni et al, 2021). The immune system itself is also increasingly understood to be a crucial variable in the tissue microenvironment composition equation (Sutherland et al, 2023).

[1]Kennedy Institute of Rheumatology, University of Oxford, Oxford, UK. [2]Institute of Infection, Immunity and Inflammation, University of Glasgow, Glasgow, UK. [3]Nuffield Department of Orthopaedics, Rheumatology and Musculoskeletal Sciences, Botnar Research Centre, University of Oxford, Nuffield Orthopaedic Centre, Headington OX3 7LD, UK. ✉E-mail: kim.midwood@kennedy.ox.ac.uk

Matrix is now known to be immune mediating, with reported effects on T-cell migration and pro-inflammatory mechano-sensing (Salmon et al, 2012; Sladitschek-Martens et al, 2022), but also immune-mediated, with roles for Th2-secreted IL-13 in defining mucus layer composition or for neutrophils in acute injury matrix remodelling (Hasnain et al, 2017; Fischer et al, 2022).

Of all the IMIDs, RA was reported to have the greatest increase in age-standardised incidence rates between 1990 and 2019 (Wu et al, 2023). It is characterised by chronic dysregulated inflammation, immune infiltration, hyperplasia of the joint synovium, and bone and cartilage destruction (Firestein and McInnes, 2017). Whilst treatments targeting canonical immune mediators and immune cells have significantly improved the standard of care for people with RA, these drugs do not work in a large proportion of patients, and patients who do respond can become refractory to these treatments over time. Moreover, whilst disease remission, defined by the European League Against RA (EULAR) as no or only minimal disease activity, with a low risk of both structural progression and functional impairment (Studenic et al, 2023) does occur, the frequency of sustained drug-free remission is low, with disease returning in 85–89% of patients after cessation of treatment (Schett et al, 2021). The composition dynamics of the cellular phenotypes in RA have been extensively studied and reported. This is particularly true of the fibroblast subsets of the inflamed synovium, typically separated into Lining Layer (LL) CD90- PRG4+ fibroblasts and Sub-lining (SL) CD90+ fibroblasts (Mizoguchi et al, 2018; Croft et al, 2019; F. Zhang et al, 2019; Wei et al, 2020). Novel therapeutic strategies aim to harness the specific roles of individual fibroblast subsets to restore tissue homeostasis (Croft et al, 2019; Pratt et al, 2021), and recent patient stratification efforts have highlighted fibroblast-dominant tissue pathotypes associated with poor treatment responses (Zhang et al, 2023). Various observations have also hinted at ECM dysregulation being an important feature in the pathology of RA. Overexpression of matrix-metalloproteinases (MMPs), proteases that degrade an array of matrix molecules, in the inflamed synovium has been long known, and more recent data points to overexpression of tenascin-C (TNC), an endogenous TLR4 agonist, being a key driver of RA chronicity (Midwood et al, 2009; Aungier et al, 2019). The extensive cellular characterisation, hints towards importance of matrix dysregulation, and the availability of recent publicly available datasets comparing disease types and activity statuses (Zhang et al, 2019; Alivernini et al, 2020; Data refs: Zhang et al, 2019; Alivernini et al, 2020) make RA a powerful model for studying ECM dysregulation in IMIDs.

Here, we established an analytical pipeline combining scRNAseq and multiplexed immunofluorescent imaging of matrix genes and proteins, into an atlas of ECM dysregulation in the RA synovium. In doing so we revealed intertwined roles for fibroblasts and myeloid cells in defining tissue composition, as well as distinct microarchitecture-defining roles for specific fibroblast subsets, with perturbations to these profiles depending on disease activity states. One such perturbation was in the synthesis and turnover of collagen VI, a beaded microfilament-forming collagen with roles in tissue homeostasis and fibrosis (Cescon et al, 2015; Williams et al, 2020). Combining transcriptomic analysis with proteomic and degradomic analysis, we demonstrate that collagen VI deposition by fibroblasts is associated with remission-linked cellular phenotypes and inflammation resolution in human and mouse synovitis,

respectively. Collagen VI was found to play a regulatory role in the synovium, preventing unrestricted immune ingress by blocking the migration of immune cells. Monocytes were unconstrained by collagen VI via their ability to actively degrade it, thereby releasing detectable fragments and expanding the boundaries of adaptive immune-permissive niches. Together, this work sheds new light on the role of myeloid-stromal interactions, via ECM regulation, in the modulation of immune ingress and the induction of inflammation resolution. These interactions, upholding a fragile balance between matrix anabolism and catabolism, are a new piece in the puzzle of dysregulated tissue microenvironments in IMIDs, and should be harnessed as novel biomarkers and exploratory therapeutic targets.

# Results

## Stromal and myeloid cells coordinate the production of the RA synovium architectural building blocks

In recent years, the increasing availability of bioinformatic datasets has empowered biomedical research towards important discoveries. The matrisome is one such dataset, combining 1027 experimentally validated and bioinformatically predicted genes encoding the building blocks of the ECM (Naba et al, 2016). The field of RA has also benefited from the development of valuable datasets, including a scRNAseq dataset compiling whole synovial cells from four patients with active RA and 3 patients with RA in remission (DAS28 < 2.6) (Alivernini et al, 2020, Data ref: Alivernini et al, 2020). To first catalogue the synovial matrisome, gene set enrichment analysis was applied to look for specific enrichments in the expression of each matrisome gene category defined by Naba and colleagues (Appendix Fig. S1), in each cell cluster present in this dataset (Fig. 1A), using the AUCell package (Aibar et al, 2017). The matrisome can be divided into two main categories, each with three subcategories: (1) core matrisome: which is important for the structural properties of the matrix and encompass glycoproteins, essential for cell-matrix interactions, collagens that provide the main tissue tensile strength and proteoglycans, characterised by the presence of highly charged glycosaminoglycans (GAGs) attached to a protein core, and (2) matrisome-associated proteins that play regulatory roles within the matrix and encompass ECM-affiliated proteins that share structural similarities with or are known to associate with core matrix proteins, ECM regulators, enzymes involved in ECM remodelling and cross-linking and secreted factors, molecules that bind to core ECM proteins and mediate signalling and communication. Strong enrichment across all three categories of core matrisome genes was observed in the fibroblast subsets (Fig. 1B–D), with PRG4+ lining fibroblasts being less enriched than sub-lining fibroblasts. Vascular and lymphatic endothelial cells (VECs and LECs) were also found to contribute to glycoprotein expression. The expression of matrix-associated genes was less solely reliant on fibroblasts, with myeloid cells, including macrophages and dendritic cell subsets, contributing to the expression of ECM-affiliated and ECM regulator genes (Fig. 1E–G). For completeness and future reference, a full breakdown of the expression of each matrisome gene in each of the synovial cell subsets was performed and is reported in (Appendix Figs. S2 and S3). These results show that whilst fibroblasts, and to a lesser extent endothelial cells, make a large

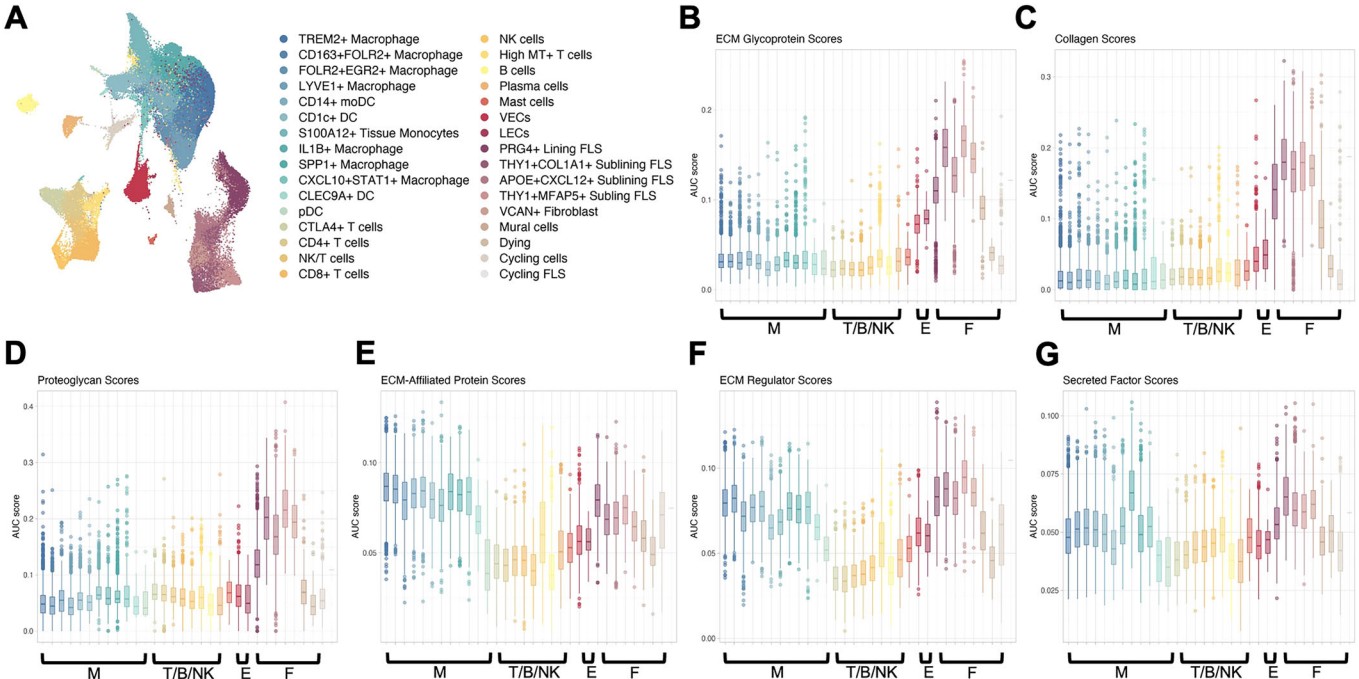

**Figure 1. Matrisomal gene set enrichment of synovial cell subsets in rheumatoid arthritis.**

The Alivernini dataset (Data ref: Alivernini et al, 2020) was plotted in UMAP space and cell subsets were assigned based on published annotations. New analysis was performed to examine matrisome subcategory expression enrichment for each of the published subsets using AUCell. (A) UMAP of total synovial cell clusters from the analysed Alivernini dataset. (B–G) Boxplots of the Glycoprotein (B), Collagen (C), Proteoglycan (D), ECM-Affiliated Protein (E), ECM Regulator (F), and Secreted Factor (G) expression enrichment scores (AUC score on the y axis) for each of the identified synovial cell subsets (x axis), as defined by AUCell. M myeloid cells, T/B/NK T-cells/B cells/Plasma cells/NK cells, E endothelial cells, F fibroblasts. The boxplots are plotted following the default geom_boxplot() function of ggplot2 and therefore represent the following statistics: lower whisker = lower whisker extends from the hinge to the smallest value at most 1.5 * IQR of the hinge; 1st Quartile or hinge (Q1) = 25th percentile; Median = 50th percentile; 3rd Quartile or hinge (Q3) = 75th percentile, upper whisker = maximal value no further than 1.5*IQR from the hinge (where IQR is the interquartile range, or distance between the first and third quartiles); outliers beyond whiskers = shown as dots. N = 20,000 randomly downsampled cells, across n = 22 patient synovial samples. Source data are available online for this figure.

proportion of core matrix proteins, other cell types also make these molecules. For example, although immune cells, including myeloid and T cells, express fewer core matrisomal genes, they do express a handful of collagens and glycoproteins at very high levels, and these may be physiologically important contributions to any local environment. Moreover, whilst macrophages express many matrix-digestive enzymes, the soluble cues that these cells make are not limited to proteases but encompass a variety of modulating enzymes including collagen cross-linkers, sulfatases and kinases. On top of this, many stromal and lymphocytic cells also participate in the synthesis of these matrisome-associated molecules. Together, these data highlight the importance of considering ECM production as a holistic co-operative venture in which all tissue-resident cells play a role.

## Spatially and functionally distinct fibroblasts define different sub-synovial niches through unique matrix expression profiles

Having identified fibroblasts as the primary matrix-defining cell type in the synovium, subsequent analyses were focused on these cells. To effectively harness past work on fibroblasts in RA, the fibroblasts from the Alivernini et al dataset were subsetted from the published and deposited raw data, and clustered from scratch to

normalise and recapitulate the four fibroblast clusters defined in the separate Accelerating Medicines Partnership (AMP) consortium study (Appendix Fig. S4), referred to throughout the rest of the study as AMP1 (Zhang et al, 2019; Data ref: Zhang et al, 2019), and enable cross-dataset comparisons. The well-documented specific gene expression profiles of the AMP1 fibroblast clusters, their currently-reported spatial distribution (confirmed in this work by pseudotime analysis, Appendix Fig. S5), broad applicability to mouse datasets (Wei et al, 2020), and recent use in a cross-tissue fibroblast atlas (Korsunsky et al, 2022), make them particularly useful analytical tools. These four cell clusters comprise three populations of fibroblasts that locate to the sub-lining layer of the synovium and participate in tissue remodelling (SC-F1, SC-F3) and inflammatory processes (SC-F2), along with one population of fibroblasts that locate to the lining layer of the synovium and participate in barrier function and synovial fluid production (SC-F4). Subset-specific transcriptomic profiles of the sub-lining SC-F1 and SC-F3, immune-associated sub-lining SC-F2 (Mizoguchi et al, 2018; Zhang et al, 2019), and lining SC-F4 (Croft et al, 2019) can therefore aid in the study of RA synovial matrix heterogeneity and uncover differential matrix composition in distinct sub-synovial niches.

In parallel to this de novo "canonical" clustering, fibroblasts were also clustered using exclusively matrisomal genes (Fig. 2A),

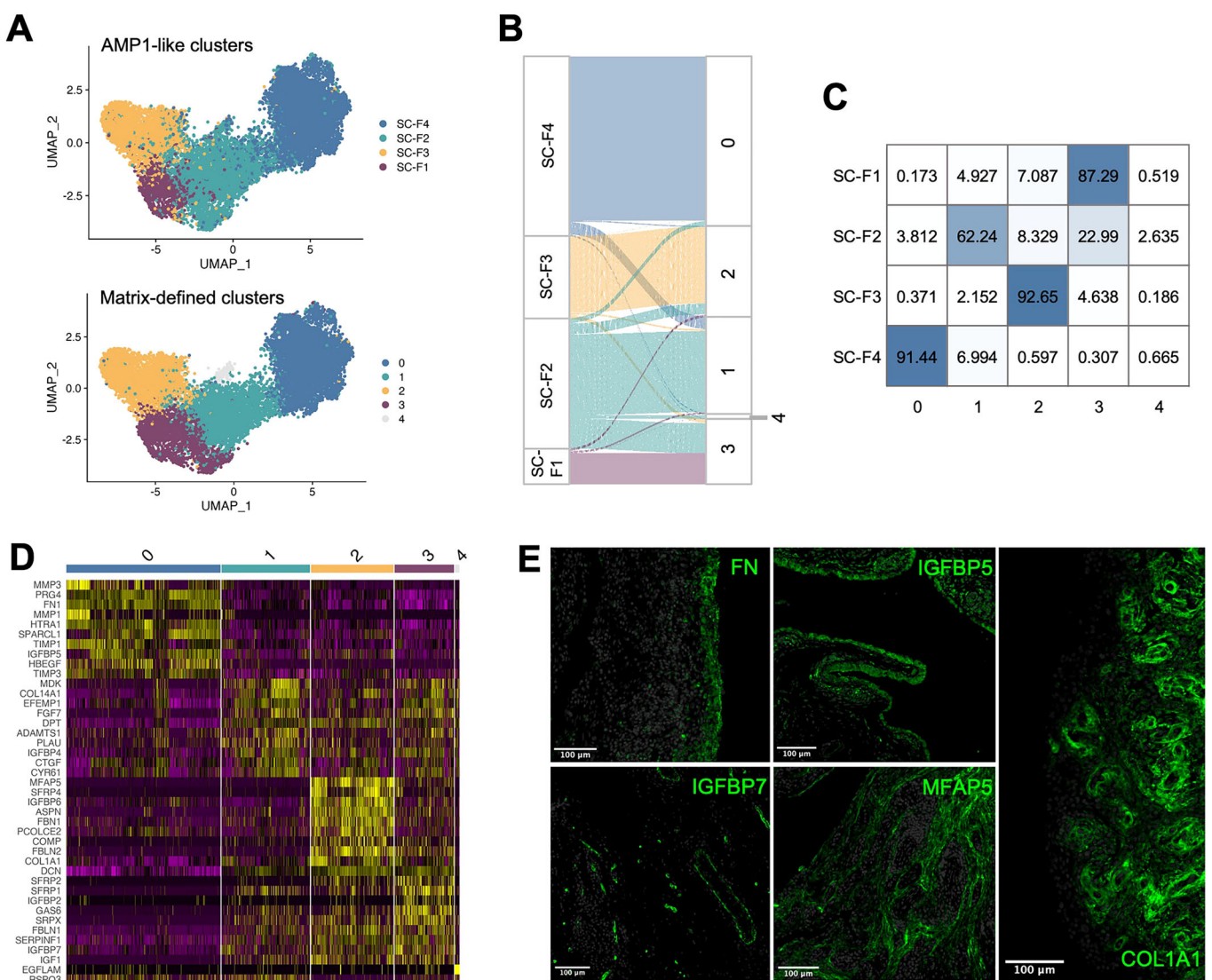

**Figure 2. Canonical fibroblast subsets are defined by distinct matrix expression profiles.**

Raw data and cells from the Alivernini dataset were analysed de novo to subset out fibroblasts as detailed in Appendix Fig. S4. Fibroblasts were subsequently reclustered into four fibroblast clusters using the Zhang dataset nomenclature (AMP1-like) (Zhang et al, 2019; Data ref: Zhang et al, 2019) to enable direct comparison of cell types across studies. Newly subsetted fibroblasts were also clustered using the matrisome gene list, excluding cytokines and chemokines, with their top cluster-defining markers chosen to investigate protein distribution in human RA synovial tissue via multiplexed immunofluorescence. (A) UMAP of the Alivernini dataset fibroblast clustered using the full available genome to redefine AMP1-like clusters (top) or using the cytokine- and chemokine-subtracted matrisome to identify matrix-defined clusters (bottom). (B) Alluvial plot of the label-transfer and contributions of synovial fibroblasts in the AMP1-like clusters to the matrix-defined clusters. (C) Table of the percentage of cells from each AMP1-like cluster that make up each of the matrix-defined clusters. (D) Heatmap of the top ten cluster-defining matrix genes for each of the matrix-defined clusters. (E) Representative immunofluorescent staining on human RA patient synovial biopsies for key fibroblast cluster and niche-defining matrix markers fibronectin (FN), IGFBP5, IGFBP7, MFAP5, and type I collagen (COL1A1). FN, MFAP5 and COL1A1 stains were performed on biopsies from $N = 3$ donors; IGFBP5 and IGFBP7 on biopsies from $N = 6$ donors. Source data are available online for this figure.

excluding cytokine and chemokines in the Secreted Factors category of the matrisome, to generate novel matrix-focused clusters. Matrisome-focused clustering was found to recapitulate the AMP1-defined clusters with striking overlap (Fig. 2B,C). A simplified equivalency shows cluster 0 containing 91.44% of the SC-F4 *PRG4*+ lining fibroblasts, cluster 1 62.24% of the SC-F2 *HLA-DR*+ sub-lining fibroblasts, cluster 2 92.65% of the SC-F3 *DKK3*+ sub-lining fibroblasts, and cluster 3 to be an expansion of 87.29% of the SC-F1 *CD34*+ sub-lining fibroblasts fuelled by the inclusion of

22.99% of the formerly SC-F2 fibroblasts (Fig. 2C). This analysis also identified a fifth smaller cluster characterised by the highly specific expression of *EGFLAM*, a gene associated with cell proliferation (Chen et al, 2019).

Differential expression analysis (DEA) between the matrix-focused clusters unveiled new and distinct matrix expression profiles associated with each of these tissue architecture-defining subsets (Fig. 2D). Cluster-defining matrix gene cassettes, combined with pathway analysis, identified lining cluster 0 as being highly

ECM regulatory, with specific expression of the proteases *MMP1/MMP3*, protease inhibitors *TIMP1/TIMP3*, while also expressing tissue repair-associated *IGFBP5* and the major structural gene *FN1* (Fig. 2D; Appendix Fig. S6). Immunofluorescent staining of RA synovial biopsies confirmed lining layer deposition of fibronectin and IGFBP5 at the protein level (Fig. 2E). Cluster 2 was characterised by the most unique of the sub-lining transcriptomic matrix signatures, with a primary pathway pointing to collagen fibril organisation, notably through *MFAP5, COL1A1, FBN1*, and *FBLN2* (Fig. 2D). This cluster defined a densely fibrillar matrix niche in tissue biopsies stained for MFAP5 and COL1A1, in the interstitial sub-lining (Fig. 2E). Clusters 1 and 3, while being most similar in their matrix expression profiles, still harboured key differences, with cluster 3 being strongly associated with negative regulation of Wnt signalling through *SFRP1/SFRP2* and the expression of major RA-associated growth factor *IGF1* (Denninger et al, 2015; Andersson et al, 2018) and its regulator *IGFBP7* (Evdokimova et al, 2012), with the latter interestingly specifically located around the vasculature, in particular co-staining with endothelial cells (Fig. 2E). Cluster 1 on the other hand, canonically defined as SC-F2 *HLA-DR+* immune-associated fibroblasts (Zhang et al, 2019), was associated with the expression of tolerogenic dendritic cell inhibitor *MDK*, and CCN genes *CYR61* and *CTGF*.

Our analysis strategy builds upon published work (Data refs: Zhang et al, 2019; Alivernini et al, 2020) to address unanswered biological questions; specifically, to provide new insights into the structure-forming functions of fibroblasts, previously unexplored in both original papers (by virtue of use of unbiased clustering strategies), and currently published work in RA. By subsetting raw data from these studies and clustering cells from scratch using matrisome genes exclusively, an approach not previously reported, novel findings were generated, which were twofold. Firstly, the identification of fibroblast matrix expression cassettes at unprecedented resolution and depth, both cluster-specific and disease activity-specific (Fig. 2D). Secondly, the observation that similar clusters are defined through matrix-biased or unbiased clustering, showing that functionally distinct fibroblast clusters also dictate the composition of distinct sub-synovial architectures (Fig. 2E).

## Fibroblasts define a cross-disease remission matrix signature

Following general characterisation of synovial niche compositions, we set out to identify novel disease-type- and disease activity-specific changes in matrix expression, to gain insights into clinically relevant tissue perturbations. To do so, we ran MAST DEA as in the original studies, and strengthened it with the addition of pseudobulked edgeR DEA, on nomenclature-matched fibroblasts from two human scRNAseq datasets (Data refs: Zhang et al, 2019; Alivernini et al, 2020) and MAST alone on an additional published mouse scRNAseq dataset (Wei et al, 2020), to look for cross-species conservation of differentially expressed genes (DEGs) (Fig. 3A). DEGs were filtered to focus specifically on matrisomal hits, and were identified for each fibroblast subset between RA and OA, or between active RA and remission, to highlight new deeper matrisomal signatures unique to RA but also matrisomal features of disease remission (Fig. 3B).

To define common and unique active inflammation and resolution signatures across diseases, we identified matrisomal DEGs following this same methodology in a dataset of human frozen shoulder (FS) (Ng et al, 2024). While different tissues, the synovium and the shoulder capsule, have similar structures, notably characterised by a joint cavity-facing lining layer and an underlying sub-lining layer (Ng et al, 2024). Disease progression is however distinct between RA and FS, with FS being self-limiting, resolving spontaneously over time. Because of this self-resolving nature of FS, we looked for overlap in DEGs between fibroblasts from the FS capsule and from the synovium of RA patients in sustained remission. In the lining layer, we identified cassettes of 10 common remission-upregulated genes (*FGL2, HTRA4, IGFBP5, IGFBP6, LTBP4, PDGFD, PRG4, S100A6, SEMA3E, SPARCL1*) and 13 common remission-downregulated genes (*ADAMTS2, ADAMTS6, COL14A1, COL1A1, COL1A2, COL3A1, IGFBP7, MMP2, MMP3, SERPINE2, SMOC1, TIMP1, TNFAIP6*) (Fig. 3C). In the sub-lining, we identified cassettes of 6 common remission-upregulated genes (*CILP, FGFBP2, FGL2, LTBP4, MGP, TNXB*) and 7 common remission-downregulated genes (*COL1A1, COL3A1, COL5A2, ELN, IGF1, SERPINE2, SPARC*) (Fig. 3D). Interestingly, we also identified a gene set unique to FS-resolution over RA remission, consisting of neutrophil chemokines *CXCL1, CXCL2*, and *CXCL8*, and monocyte attracting *CXCL3* (Fig. 3C).

These new matrisome-focused cassettes of genes point to opposing tissue landscapes based on inflammatory activity. Key features of active inflammation, that were downregulated in remission, are ECM remodelling and wound-healing responses mediated through the upregulation of ECM regulators such as *ADAMTS2, ADAMTS6, MMP2, MMP3, SERPINE2, TIMP1*, and fibril-forming collagens I and III, which point to the activation of TGF-β signalling pathways (Wang et al, 2003; Kim et al, 2004; Gomes et al, 2012; Park et al, 2015; Li et al, 2016; Cain et al, 2022). IGF signalling is also a significant part of this signature through *IGF1* and *IGFBP7*, while antigen presentation promotion has been associated with the latter (Akiel et al, 2017). Tissue-level remission on the other hand is characterised by the opposite signature, notably TGF-β and IGF inhibition through *LTBP4, HTRA4, CILP, IGFBP5*, and *IGFBP6* (Johnson et al, 2003; Heydemann et al, 2009; Ding and Wu, 2018; Wang and Nie, 2021). Inhibition of antigen presentation is also a feature of the remission cassette, through upregulation of FGL2 (Yan et al, 2019). Together, this analysis identifies a new gene expression signature that is common to remission of both RA and frozen shoulder, as well as genes that are specific to remission in both diseases. We also identify genes that are common to the active phase of both RA and frozen shoulder, as well as genes specific to each active disease. This common remission signature opens the door for a better understanding of tissue-agnostic processes that are linked with well-controlled disease, and which could lead to new ways to restore tissue homeostasis in a disease and tissue-independent manner. On the other hand, genes identified as unique to the active stage and that are different between RA and frozen shoulder disease could be useful as potential biomarkers.

These remission and inflammation resolution cassettes are of significant value in the quest for curative solutions in IMIDs. We therefore sought to investigate further specific dynamic dysregulated features of RA between active inflammation and remission. To do so, we first applied a target prioritisation scheme to our matrisomal DEG hits based on differential expression in active RA or remission in the Alivernini dataset, and species conservation in

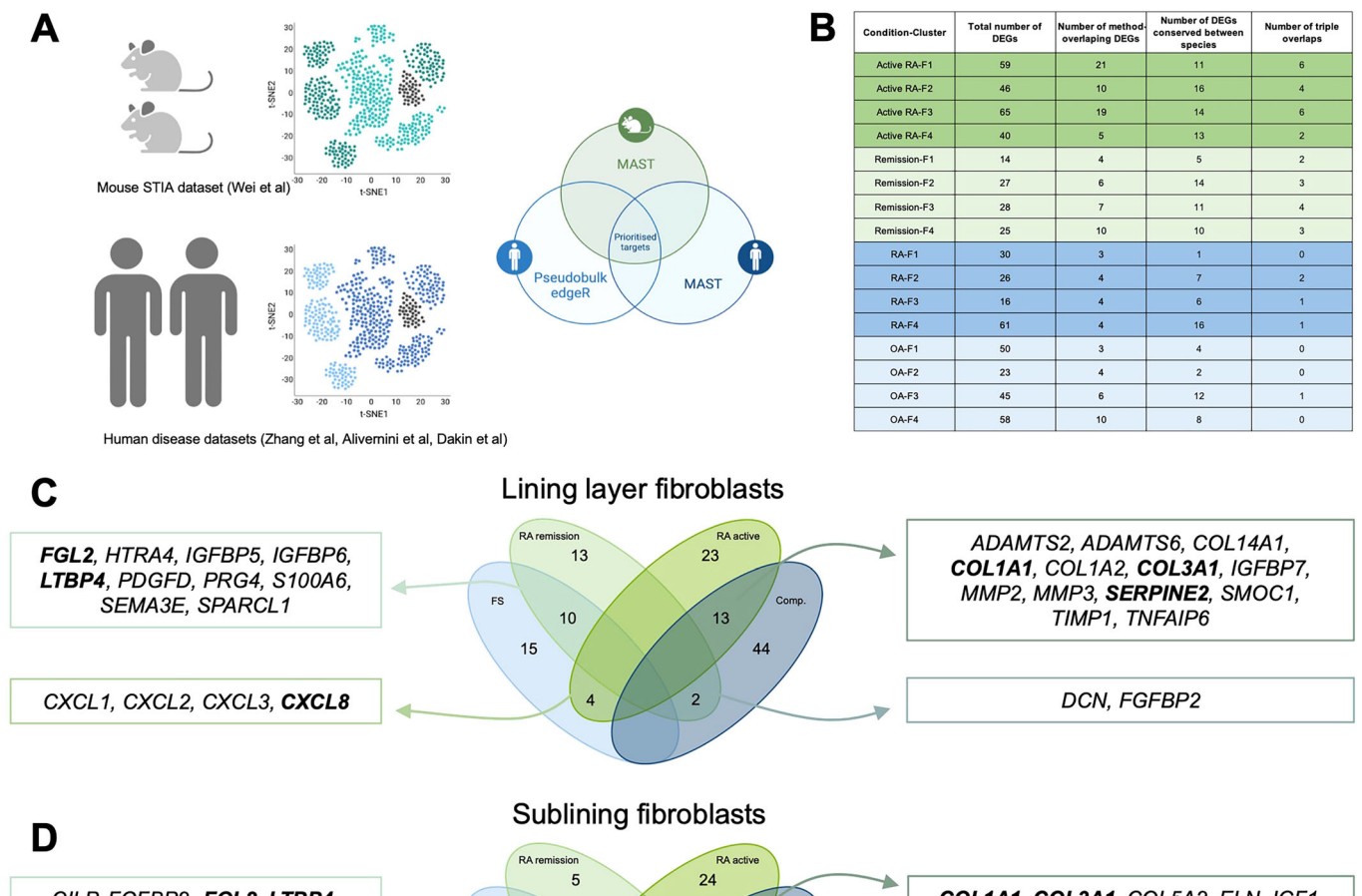

**Figure 3.  Unique and common matrisomal differential expression signatures between rheumatoid arthritis and frozen shoulder.**

Differential expression analysis between newly defined AMP1-like fibroblast clusters of the Alivernini dataset and fibroblasts from Frozen Shoulder was performed to obtain a novel list of disease activity-associated DEGs. These DEGs were compared to obtain overlapping and unique signatures between active RA and comparator tissue, and RA in remission and self-resolving inflamed frozen shoulder tissue. (A) Schematic summary of the methodology used for differentially expressed target prioritisation. (B) Table summary of matrisomal DEGs for each AMP1-like cluster of fibroblasts in both the Alivernini and AMP1 datasets in each of the target prioritisation strata. These are broken down into total DEGs, "method-overlapping" genes classified as DEGs through both pseudobulked EdgeR and MAST, species-conserved genes classified as DEGs by at least one of the two methods in both mouse and human datasets, and "triple overlap" genes classified as DEGs by both DEA methods and across species. (C, D) Venn diagrams of overlapping DEGs between active RA/remission fibroblasts and frozen shoulder/comparator capsular fibroblasts. The comparison is separated between lining layer fibroblasts (C) and sub-lining fibroblasts (D). Genes conserved between lining and sub-lining fibroblasts are highlighted in bold. Source data are available online for this figure.

the serum transfer arthritis mouse model of the Wei et al dataset for tractability.

## Collagen VI is overexpressed yet less abundant at the protein level in RA

Target prioritisation yielded a top ten list of matrisomal genes of interest in lining and sub-lining fibroblasts in active RA and RA in remission (Fig. 4A). Amongst these hits, numerous collagens, including *COL1A1, COL1A2, COL3A1, COL6A3,* and *COL14A1*

were significantly upregulated in active RA compared to remission (Appendix Fig. S7A,C,E,G). Analysis of RA versus OA tissue highlighted similar trends for higher COL1, COL3 and COL6 genes in RA compared to OA, with significantly higher COL1A1, COL6A1 and COL6A2 in RA, whilst COL14A1 expression was significantly lower in RA compared to OA (Appendix Fig. S7B,D,F,H). Interrogation of a larger integrated data set, comprising 88 RA patients and 10 OA patients (Thomas et al, 2024; Data ref: Thomas et al, 2024), and data from 13 healthy individuals (Faust et al, 2024; Data ref: Faust et al, 2024), showed higher COL1,

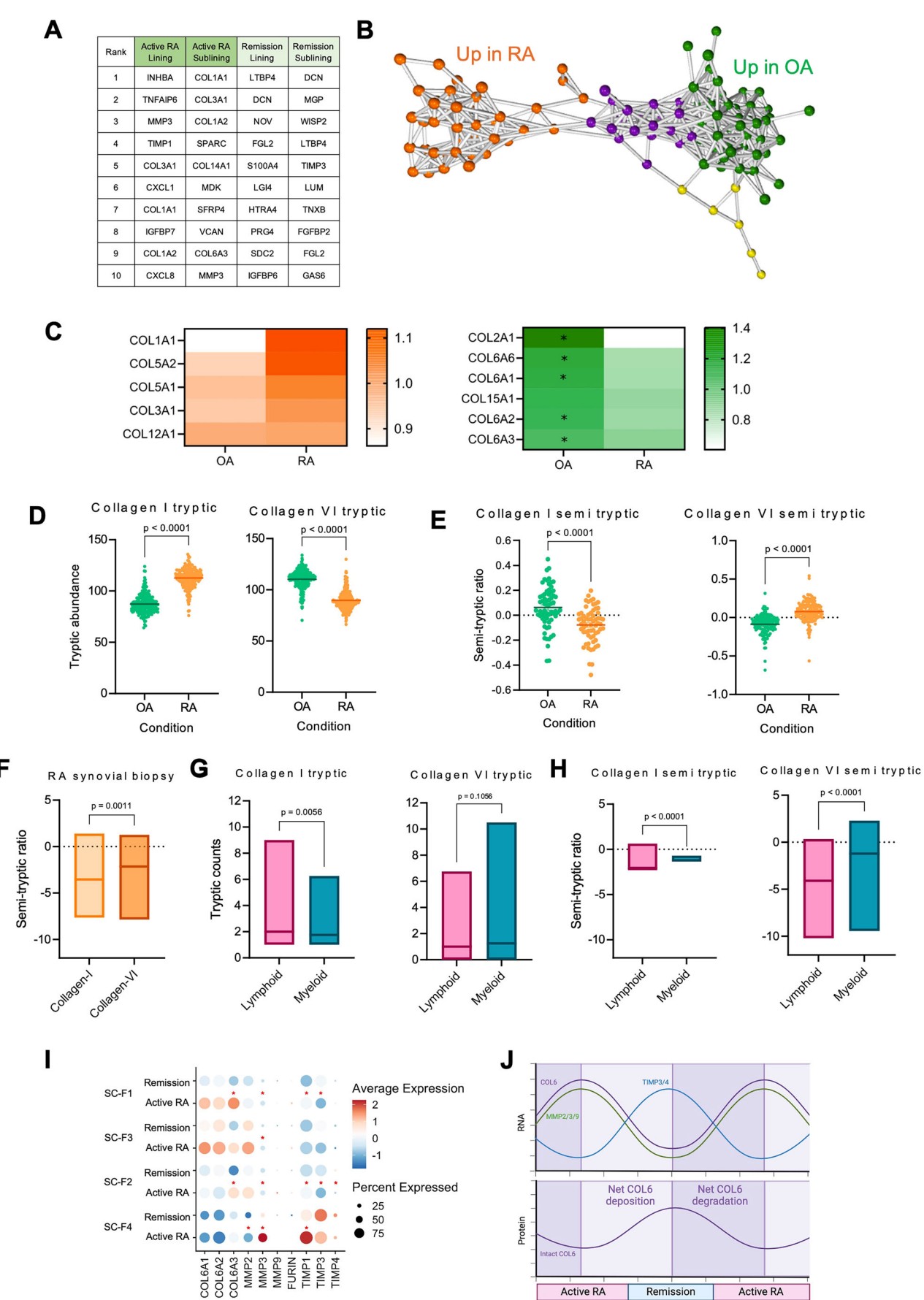

**Figure 4. Expression dysregulation of collagen VI and its processing machinery is a key feature of Rheumatoid Arthritis disease activity.**

Following differential expression analysis between newly defined AMP1-like fibroblast clusters of the Alivernini dataset, DEGs were prioritised based on DEG conservation between differential expression analysis methods edgeR and MAST, conservation in a newly performed MAST analysis of the Wei mouse dataset of STIA, log2FC and adjusted *P* values. (A) Ranks of the top ten prioritised matrisomal DEGs in active RA/remission lining and sub-lining fibroblasts. (B) Network analysis of protein abundances and correlations in Graphia. Each protein is represented by a dot. Protein clusters are grouped by colour, with orange proteins corresponding to proteins more abundant in RA, green proteins more abundant in OA, and purple and yellow proteins in neither. (*N* = 12 for OA and *N* = 10 for RA). (C) Heatmaps of protein abundance for proteins found to be overrepresented in RA (orange) and OA (green) samples. Stars denote a *P* value < 0.05 through *t* test. (*N* = 12 for OA and 10 for RA). (D) Abundance of Collagen-I and VI protein in OA and RA patient samples (*N* = 12 and *N* = 10, respectively), each dot represents the abundance of a tryptic peptide. For Collagen-I chains Col1a1 and Col1a2 have been grouped. For Collagen-VI, chains Col6a1, Col6a2 and Col6a3 have been grouped. (E) The ratio of semi-tryptic collagen-I peptide abundance compared to the average tryptic peptide abundance, this is as an indicator of degradation for Collagen-I and -VI in OA and RA. Each dot represents the ratio for a single semi-tryptic peptide. (*N* = 12 for OA and 10 for RA). (F) Supporting data from RA synovial biopsy (*N* = 8) dataset. Semi-tryptic ratio of collagen-I and –VI based on relevant spectral counts. Tryptic spectral counts (G) and semi-tryptic ratio (H) from the same dataset as (F) but separated into lymphoid (*N* = 4) and myeloid (*N* = 4) endotypes for grouped collagen-I and –VI. *P* values for *t* tests are labelled above each dot plot. (I) Dot plot of genes for type VI collagen chains and its processing regulators in each AMP1-like fibroblast cluster, split by disease activity status, coloured by scaled average expression, and sized by percentage of expressing cells. Red asterisks separate comparisons that led to genes being classified as DEGs. (J) Suggested mathematical model for out-of-phase oscillations in MMP2/MMP9 and TIMP3/TIMP4 expression, the subsequent anabolism potential of the tissue they induce, and its putative impact on collagen VI deposition. (F: collagen I vs collagen VI in RA synovium *P* = 0.0011, G: lymphoid vs myeloid counts for collagen I *P* = 0.056 and for collagen VI *P* = 0.1056). Exact *P* values are given unless they are below 0.0001 which is beyond the calculation limit of the software used. Source data are available online for this figure.

COL3, COL6 and COL14 in normal synovium compared to both OA and RA synovium, but no significant difference between disease status (Appendix Fig. S7I). Pseudobulk analysis of collagen genes subsetted by patient demonstrated highly variable expression between individuals with RA; and this was observed across all fibroblast subsets (Appendix Fig S7J shows COL6A1 as an exemplar gene). This large dataset includes samples taken from different joint sites as well as different cell-type abundance phenotypes (CTAP) identified by Zhang et al (Zhang et al, 2023; Data ref: Zhang et al, 2023), where RA patients were clustered into groups based on their cellular abundance, giving rise to 6 CTAP groups: endothelial-fibroblast-myeloid (EFM), fibroblast (F), T-cell-fibroblast (TF), T-cell-B-cell (TB), T-cell-myeloid (TM) and myeloid (M). Linear regression analysis on pseudobulked data for each fibroblast cluster show that both site and CTAP affect collagen gene expression in RA patients. In general, across all clusters, samples acquired from the ankle appear to have lower expression of collagens, compared to knees, fingers and toes which display increased expression, whilst CTAP:F showed lower collagen expression with a trend for CTAP:M having the highest expression (compared to the reference CTAP:EFM) (Appendix Fig. S7K,L). Considering the effect of site and CTAP on collagen expression, we used edgeR to run a generalised linear model likelihood ratio test (glmLRT), to formally test whether expression of collagens significantly differ between OA patients and each RA CTAP, within a single joint site (knee). Although numbers of patients of each CTAP are small (*n* = 3 CTAP:T + F, *n* = 5 all other CTAP) there is a clear trend showing increased expression of collagens 1, 3 and 6, but not COL14, in lining fibroblasts of RA patients with a CTAP:M phenotype, and COL6A1 was significantly upregulated between RA CTAP:M and OA patients (Appendix Fig. S7M). Together, these data identify a subset of collagen genes (COL1, COL3, COL6, COL14) whose expression is increased in active RA compared to RA in remission and healthy synovium. Expression of each of these collagens is also high in OA synovium compared to healthy tissue, with more COL1 and COL6, and less COL14, in RA compared to OA, and with COL6A1 elevated in particular in RA patients with myeloid-rich disease.

To compare transcriptomic data with protein expression, a proteomic dataset of human OA and RA synovium (PRIDE, project PXD027703) was analysed to look for ECM protein abundance. Protein clusters were formed based on their upregulation in OA or RA and their covariance (Fig. 4B). Amongst these, a number of collagens were also identified. COL1A1, COL5A1/2, COL3A1, and COL12A1 were found to be marginally more abundant in RA compared to OA, whilst COL2A1 and COL6A1/2/3/6 were significantly more abundant in OA compared to RA (Fig. 4C). The overabundance of type I and type III collagen proteins in RA compared to OA mirrors mRNA levels of COL1A1 and COL3A1. However, type VI collagen followed an opposite pattern, with protein less abundant in RA compared to OA, whilst mRNA levels were higher in RA. To explain this dichotomy, we explored ECM turnover as a potential culprit, comparing the abundance of tryptic peptides (a measure of protein abundance, peptides released during tissue digestion with trypsin for LC-MS/MS) and semi-tryptic peptides (used here as a measure of degradation, peptides released by endogenous proteases). A higher ratio points to increased levels of degradation in the tissue, while a lower ratio points to decreased levels of degradation. Collagen I exhibited significantly lower semi-tryptic to tryptic ratios in RA compared to OA, whilst type VI collagen had significantly greater semi-tryptic to tryptic ratios in RA compared to OA (Fig. 4D,E). This finding held true when investigating individual collagen chains and patient groups (Appendix Fig. S8A–D). To validate this disease specificity in collagen turnover, we analysed OA synovial fluid and RA synovial biopsy tissue from two independent datasets in a similar way (PXD016620 and PXD020397). Semi-tryptic peptide counts, indicative of degradation, were higher for collagen I than collagen VI in OA synovial fluid, whilst the semi-tryptic to tryptic ratio for collagen VI was significantly higher in RA, supporting our previous analysis (Appendix Fig. S8E,F; Fig. 4F). The latter dataset comprised RA tissue from two different, histologically defined, disease endotypes; myeloid and lymphoid (Lewis et al, 2019). Subsetting this analysis based on patient endotype revealed collagen I had a significantly lower semi-tryptic to tryptic ratio in myeloid compared to lymphoid, whilst type VI collagen had significantly greater semi-tryptic to tryptic ratios in myeloid compared to lymphoid (Fig. 4G,H). These data highlight increased collagen VI protein degradation in RA, and in particular myeloid-rich disease, an effect that is not observed for collagen I.

With these new insights, we examined expression of the published ECM regulators of collagen VI in a transcriptional model of collagen VI dynamics and RA disease activity (Bourboulia and Stetler-Stevenson, 2010; Veidal et al, 2011). RA was characterised by higher relative expression of proteases *MMP2* and *MMP9*, enzymes known to cleave collagen VI, with a comparative decrease in OA (Appendix Fig. S9). The opposite expression distribution was observed for their specific inhibitors, with *TIMP3/TIMP4* being higher in OA and lower in RA (Appendix Fig. S9), supporting experimental data showing elevated degradation of this collagen in RA tissue. These data suggest that whilst collagen VI mRNA levels are higher in both disease states compared to healthy tissue, different protein levels in RA and OA are due to altered protein turnover. Similar patterns of expression collagen VI regulators were observed when comparing active RA and RA in remission, with higher collagen VI degrading enzymes and lower enzyme inhibitors in active RA (Fig. 4I). We then modelled the clinically-characteristic cycling between active RA and remission phases as out-of-phase waves to model putative conversion of transcribed *COL6* into deposited protein levels (Fig. 4J). This revealed active RA to be a disease stage characterised by high collagen VI degradation potential, with many active proteases, and remission to be characterised by higher deposition potential, with fewer proteases and large amounts of their inhibitors (Fig. 4H,I).

Together, these data show dysregulation of both mRNA and protein level expression of collagen VI, with transcription and protein degradation elevated in RA, particularly in myeloid-rich disease, through altered expression of its established regulators. These findings exemplify that whilst understanding patterns of gene expression at mRNA level provides valuable information revealing which genes are turned on and in which cell types, these data do not always translate into changes in tissue protein levels, particularly for matrix molecules (Nieuwenhuis et al, 2021). Complex post-transcriptional and post-translational modifications that these molecules are subject to, which ultimately define protein abundance, also occur in a tissue and disease-specific manner. Here, by combining transcriptomic analysis with proteomic and degradomic analysis, we start to provide insight into the functional consequences of mRNA induction coupled with low or high levels of protein breakdown, highlighting how turnover of specific collagens dictates tissue matrix levels.

## Type VI collagen accumulation correlates with markers of remission in human and mouse synovial inflammation

To begin to dissect the functional implications of type VI collagen production in the synovium, we performed an interactome analysis on COL6 genes using Matrix DB. This identified 457 known or predicted binding partners for COL6A1, A2 and A3 (Table EV1). Pathway analysis revealed enrichment of genes involved in matrix remodelling, immune cell migration, cytoskeletal-mediated cell motility, cell survival and proliferation, and immunoregulatory interactions between lymphoid and non-lymphoid cells (Appendix Fig. S10A). StringDB interaction analysis filtered for the highest stringency confidence, and experimentally validated physical interactions, identified 158 COL6 binders that clustered into 28 different networks (Appendix Fig. S10B). In line with known roles for COL6 in ECM molecule binding, matrix remodelling and

fibrosis (Williams et al, 2020), just under one third (28%) of all binders were matrisomal genes, and 5 of the 28 clusters were linked to basement membrane matrix, matrix remodelling (including proteases MMP2 and MMP9) and TGFβ signalling, which comprised the largest cluster. Notably, 30% of all binders were plasma membrane-associated proteins (Appendix Fig. S10C). mRNA expression of each of these 47 cell surface receptors was detected in the synovium in two independent single-cell RNA sequencing datasets, with some expressed in multiple cell types, and others restricted to different immune cell populations or to stromal cells (Appendix Figs. S11 and 12). Amongst the synovial cell receptors for COL6, several were regulated across disease status, for example, expression of CD2 and CD48 was elevated in T cells in active RA compared to remission, whilst CD47 and ICAM3 were elevated in T cells in remission compared to active RA. In the myeloid compartment integrin αX was elevated in subsets of DCs, macrophages and monocytes in active RA, whilst integrin β7 was elevated in DCs and T-cell subsets in remission (Appendix Fig. S12). Mapping the synovial target cells that express COL6 binding partners therefore reveals that, as well as acting as a regulator of matrix remodelling, altered tissue levels of COL6 likely impacts myeloid and lymphocytic cell behaviour by virtue of direct binding to these cell types, and that differential expression of COL6 binding partners across disease status may shift how this collagen signals to cells in distinct phases of synovial inflammation.

We next examined the spatial distribution of COL6A1 in the synovium in the context of these immune cell types via multiplexed immunofluorescence on synovial biopsies from RA and OA patients. Using a panel of stromal and immune markers, the lining layer (LL), interstitial sub-lining (iSL), and lymphoid aggregate (LA) niches of each section were identified, and their cellular composition characterised (Appendix Fig. S13). Perivascular areas of each tissue section were automatically annotated by defining a region with 50 μm radius out from individual COL4A1+ vasculature as their centroid, and the percentage of COL6A1+ perivascular area was measured. To gauge potential immunological associations, correlations between the percentage of COL6A1+ coverage and immune cell subset composition of individual sections were performed. The strongest association between total COL6A1+ synovial area and immune cell numbers present in the tissue was observed with MerTK+ macrophages and T cells, although these were not significant. However, a significant positive correlation was observed between the COL6A1+ perivascular area and the combined proportion of MerTK+ macrophages and T cells (Benjamini–Hochberg adjusted $P = 0.029$), with a trend towards significance for MerTK+ macrophages alone (Benjamini–Hochberg adjusted $P = 0.055$) (Appendix Fig. S14). The correlation between MerTK+ macrophages was observed in tissue from both RA (Fig. 5A,D) and OA (Fig. 5B,E) synovium. These data confirm induction of collagen VI expression in both synovial disease states and indicate that the correlation between deposition in the perivascular space and macrophage status is well conserved regardless of disease type. Importantly, MerTK+ macrophages are a defining feature and hypothesised driver of remission in RA (Alivernini et al, 2020). Localisation data are consistent with published studies showing T-cell interactions with COL6 (Lin et al, 2022), and both T cell and myeloid receptors identified in our interactome analysis, and show for the first time a link between COL6 and remission-associated macrophages. Moreover, these

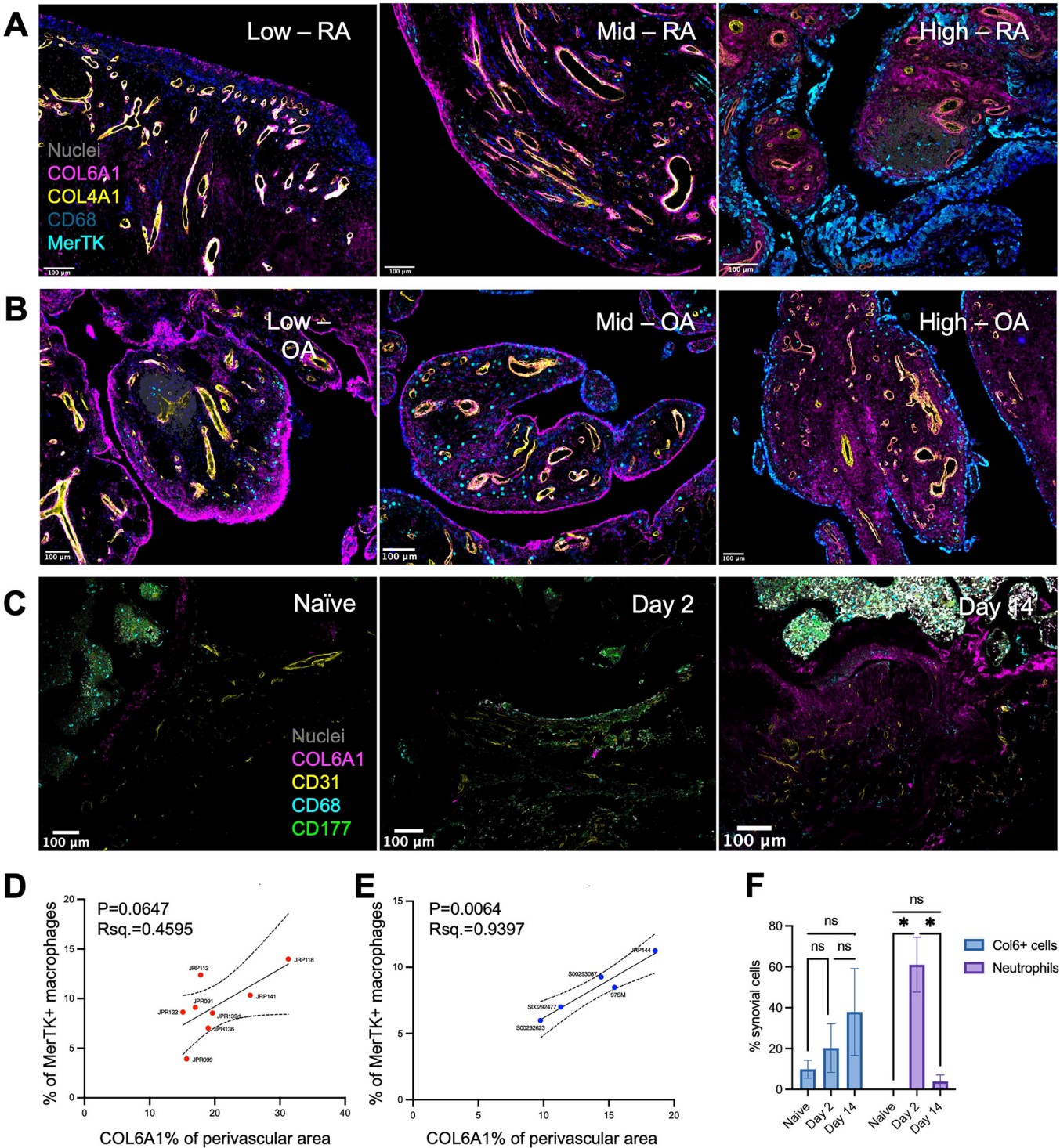

results indicate that regional deposition of COL6 is specifically associated with the cellular status of the tissue, and that this occurs in a disease-agnostic manner.

A significant difficulty surrounding the study of chronic inflammatory diseases is that of the chronology of phenomena observed in human tissue. To address this, sections from mouse knees having undergone the antigen-induced arthritis (AIA) model

of synovitis were stained for COL6A1, CD31[+] vasculature, CD68[+] macrophages, and CD177[+] neutrophils. Sections from naïve mice, day 2 post-induction (peak inflammation), and day 14 post-induction (complete resolution) were chosen to define the chronology ranging from homeostatic healthy synovium, to peak primary inflammatory insult, and post-inflammation resolution and tissue repair (Fig. 5C). As previously described in the literature

◀

**Figure 5.  Perivascular collagen VI deposition associates with inflammation resolution in human RA, human OA, and mouse antigen-induced arthritis.**

(A) Multiplexed immunofluorescent staining images depicting regions of interest from three separate RA biopsies. The leftmost image is from a section with low perivascular COL6A1 and low MerTK+ macrophage tissue prevalence, the middle image from a section with intermediate levels of both metrics, and the rightmost from a section with high levels of both metrics. (B) Multiplexed immunofluorescent staining images depicting regions of interest from three separate OA biopsies, following the same display pattern as (A). (C) Confocal immunofluorescent images of the synovial tissue of representative naive mice, mice on day 2 post AIA induction, and mice on day 14 post AIA induction. Representative images were selected from $N = 8$ RA sections, $N = 5$ OA sections, and $N = 3$ mice for each time point. (D) Correlation between MerTK+ macrophages as a percentage of all detected tissue cells and COL6A1+ area as a percentage of total perivascular area in RA sections. Correlations were run in PRISM, Rsq=Pearson correlation coefficient, $P = P$ value from two-tailed test, Benjamini–Hochberg adjusted. (E) Correlation between MerTK+ macrophages as a percentage of all detected tissue cells and COL6A1+ area as a percentage of total perivascular area in OA sections. Dotted lines represent the 95% confidence interval of the linear regression applied to the data. Correlations were run in PRISM, Rsq=Pearson correlation coefficient, $P = P$ value from two-tailed test, Benjamini–Hochberg adjusted. (F) Percentage of synovial cells classified as neutrophils or COL6A1+ cell in each stained section for each timepoint. Mouse timepoint statistical test is a two-way ANOVA with multiple parameters testing via Tukey test (*$P < 0.05$, Naive vs Day 2 $P = 0.0214$, Day 2 vs Day 14 $P = 0.0615$). Source data are available online for this figure.

(Zec et al, 2023), neutrophils in tissue were absent in naive tissue, peaked on day 2, and returned to near-baseline on day 14. In line with published data in mice, COL6A1 levels were very low in naive synovium, increasing slightly on day 2, and at their highest on day 14 (Fig. 5F). This observation is reminiscent of the COL6A1 accumulation dynamics in human synovial biopsies (Fig. 5A,B,D,E), with the greatest levels of accumulation correlating with remission-associated cellular phenotypes. Taken together with transcriptomic/proteomic signatures and collagen VI deposition potential (Fig. 4), imaging in human synovia and mouse models of synovial inflammation highlights collagen VI protein accumulation in the perivascular niche as a feature of remission-like disease phases.

## Type VI collagen "dark" zones delineate synovial immune hubs

To investigate in more detail the geography of type VI collagen deposition in synovial tissue, the spatial distribution of COL6A1 throughout sub-synovial niches was examined further. In doing so, pockets of tissue devoid of COL6A1 staining were discovered across sections, in both RA and OA patient samples, and termed COL6A1 "dark" zones. These "dark" zones were found to be the exclusive sites of immune cell aggregates, regardless of the size of the aggregate, ranging from small accumulations of cells to large Tertiary Lymphoid Structure (TLS)-like aggregates (Fig. 6A,B). Strikingly, similar "dark" zones and immune restriction patterns were found in inflammatory bowel disease (IBD) and head and neck squamous cell carcinoma sections, suggesting the existence of these "dark" zones are a cross-disease and cross-tissue feature (Appendix Fig. S15). Collagen VI "dark" zone deposition patterns were also found to be drastically different to other collagens such as collagen I and IV, suggesting some degree of specificity of this effect (Appendix Fig. S16). Collagen VI-rich and -poor zones were further mapped in three OA and RA sections through local fibre density analyses, and preferential cell subset localisation was calculated through Fisher's exact test, revealing a broad spectrum of immune confinement (Fig. 6C). B cells and CD4+ T cells were found to be associated with COL6A1-poor areas in four and five out of the six sections, respectively. CD8+ T cells were more nuanced, being significantly enriched in COL6A1-poor and COL6A1-rich regions in two sections each, with no statistically significant enrichments in the other 2. Dendritic cells mirrored the distributions seen in both CD4+ T cells and B cells, with a clear enrichment in COL6A1-poor regions in four out of six sections. Plasmablast distribution was

highly variable, with a slight predilection for COL6A1-rich regions. Neutrophils were characterised by high inter-patient variability, with significant COL6A1-poor enrichment in three sections and COL6A1-rich enrichment in two sections. Macrophages were predominantly found in COL6A1-poor areas. This effect was however stronger in OA sections than RA sections, which could be due to greater sub-lining macrophage expansion in RA. Monocytes on the other hand were found to be the only cell-type significantly enriched in COL6A1-rich regions across all sections.

Based on the spatial distribution of COL6, the apparent immune retention in COL6-poor niches, and the enrichment of COL6 interactors linked with immune infiltration/cell migration pathways, we hypothesised that deposited collagen VI inhibits immune cell migration, thereby containing immune cells in the "COL6 dark" zones. To test this hypothesis, the migratory activity of monocytic THP1 cells and Jurkat T-cells on fibronectin (FN)-coated or collagen VI-coated transwell inserts was measured. Both cell lines migrated effectively on FN-coated inserts towards a FBS chemotactic gradient (FN 0.5/10), but not on collagen VI-coated inserts (COLVI 0.5/10), where migration was equivalent to non-directed cell movement across a FN-coated insert lacking any chemotactic gradient (FN 0.5/0.5). Moreover, collagen VI significantly inhibited cell migration on FN substrates (FN:COLVI 0.5/10) in a dose dependent manner (Fig. 6D). This suggests immune cells may be restricted to "COL6 dark" zones (a) because they cannot use collagen VI itself as a migratory substrate and (b) because collagen VI prevents inhibition of cell movement on migration-competent substrates found at high levels in synovial stroma such as FN.

## Monocyte infiltration releases specific detectable collagen VI fragments

Data so far show that most immune cells are spatially restricted within "COL6 dark" zones in the synovium, with the exception of CD14+ monocytes which exhibited near-free tissue roaming. Tissue-infiltrating primary monocytes are proteolytically competent (Van Goethem et al, 2009), and serve as a major source of MMP2 and MMP9 in the RA joint (Zhou et al, 2014), implicating these cells as potential mediators of collagen VI turnover in the perivascular space. To assess this, we looked at colocalisation between CD14, COL6A1, and collagen hybridising peptide (CHP) staining. CHP is a fluorophore-conjugated peptide with strong affinity for unfolded triple helical domains of collagens (Li et al,

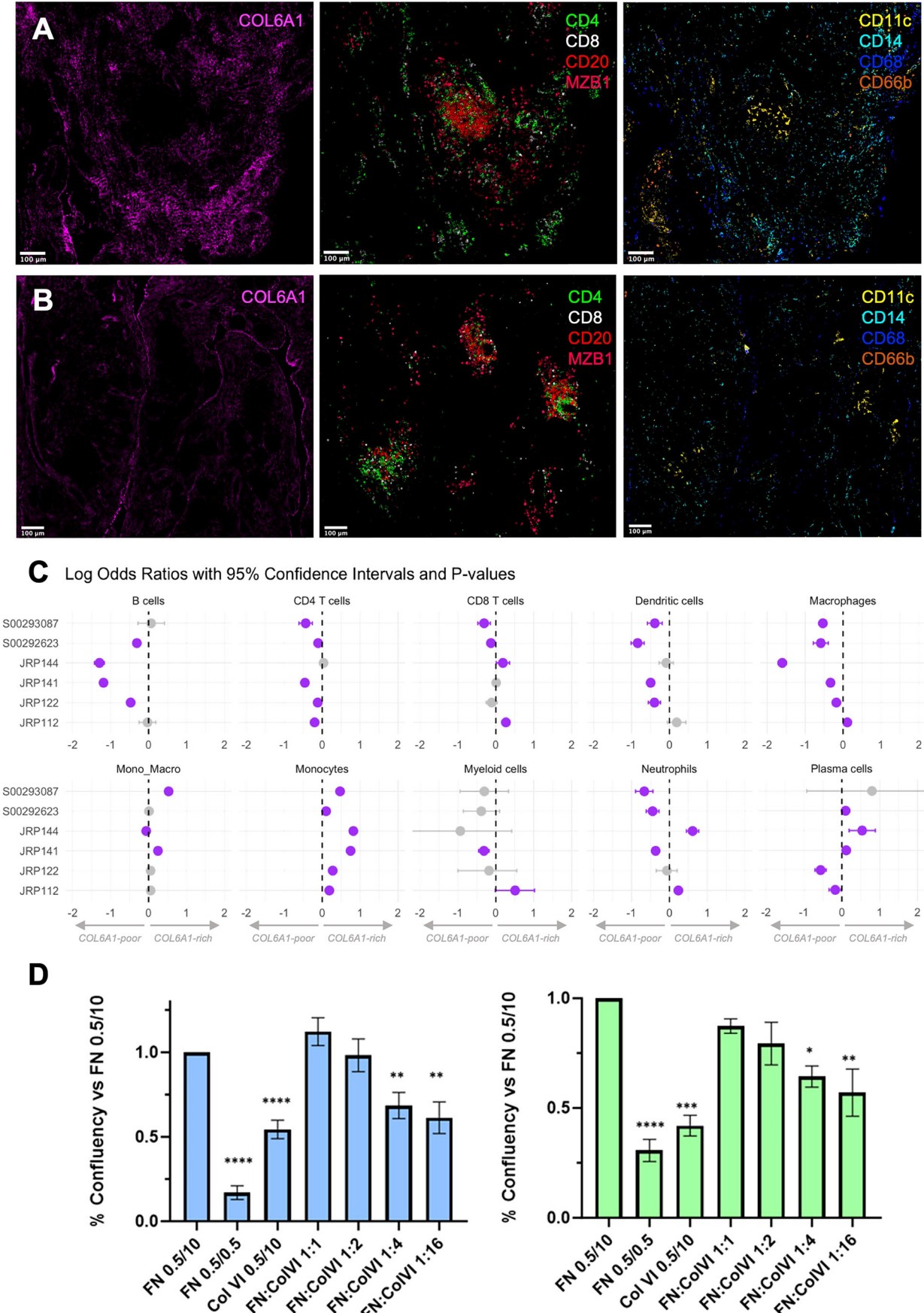

◀ **Figure 6. Collagen VI deposition patterns create "dark" zones which restrict tissue immune ingress.**

(A, B) Visualisation of COL6A1 and its dark zones, lymphocyte markers CD4, CD8 (T cells), CD20 (B cells), and MZB1 (plasma cells) and myeloid and granulocyte markers CD11c (synovial dendritic cells), CD14 (monocytes), CD68 (macrophages), and CD66 (neutrophils) in RA (A) and OA (B). (C) Dot plots of log odds ratios from Fisher's exact test of cell frequencies in COL6A1-rich regions compared to background tissue. Log odds ratios are presented for each cell type identified and each analysed section. Greyed out points represent tests that yielded a P value > 0.05, while purple points represent statistically significant data points. Error bars are the 95% confidence intervals of the test. Negative log odds ratios represent enrichment in COL6A1-poor regions, while positive ratios represent enrichment in COL6A1-rich regions. (D) Percentage confluency of THP-1 (left, blue) and Jurkat (right, green) cells that have migrated across fibronectin (FN) or collagen VI (COLVI)-coated transwell inserts, or across inserts coated with a mixture of both, with increasing concentrations of collagen VI. In all conditions cells were added to the top chamber in 0.5% FBS and migrated towards 10% FBS in the bottom chamber (0.5/10) apart from column 2 in which both upper and lower chamber contained 0.5% FBS. Data were normalised to FN (0.5% FBS/10% FBS) and analysed via a one-way ANOVA and Sidak's multiple comparisons test in GraphPad prism 10 (*$P < 0.05$, **$P < 0.01$, ***$P < 0.001$, ****$P < 0.0001$) and is represented as mean $+/-$ SEM from at least three independent experiments. Exact adjusted P values are reported for the following comparisons to FN 0.5/10 in THP1 cells: FN 0.5/0.5 $P < 0.0001$, ColVI 0.5/10 $P < 0.0001$, FN:ColVI 1:4 $P = 0.0056$, FN:ColVI 1:16 $P = 0.0018$; and in Jurkat cells: FN 0.5/0.5 $P < 0.0001$, ColVI 0.5/10 $P = 0.0001$, FN:ColVI 1:4 $P = 0.0235$, FN:ColVI 1:16 $P = 0.0021$. Source data are available online for this figure.

2012, 2013; Hwang et al, 2017; Zitnay et al, 2017). The peptide binds to and labels collagen strands undergoing active degradation by proteases. CHP tissue distribution revealed the presence of rings of active degradation at the interface between collagen VI-dense interstitial sub-lining regions and "COL6 dark" zones (Fig. 7A). This may suggest the "dark" zones are actively remodelled niches, with proteolytically active cell subsets likely responsible for delineating their borders, notably CD14+ monocytes. CHP staining was also bright throughout COL6A1-rich regions, labelling patches and trajectory-like lines of active degradation. COL6A1-rich area infiltrating monocytes were found to localise with the CHP-rich trajectories and patches, pointing to an association between collagen degradation and monocyte infiltration (Fig. 7A).

To implicate monocytes in degrading collagen VI in RA, we used proteomics to map peptide fragments to their proteins of origin, and particularly collagen VI, looking for patterns of degradation in RA and OA (Fig. 7B–D). Figure 7 presents the abundance of degradation peptides mapped to specific 50 or 100 amino acid sequence sections of the COL6A1/2/3 protein structures, represented here as a bar chart. Green heatmaps show the frequency of the two canonical processors of collagen, MMP2 and MMP9, protease susceptibility sites in each section, calculated using the Manchester Proteome Susceptibility Calculator (MPSC) (Veidal et al, 2011). The sequence mapping revealed that few of the released and detectable degradation products stem from helical domains. This is particularly true for COL6A1 and COL6A2 (Fig. 7B,C), while some fragments are released from the triple helical domain of COL6A3, particularly the N-terminal part of the domain (Fig. 7D). RA samples were broadly richer in upregulated degradation products than OA samples across the three chains of collagen VI. While many of the released fragments were upregulated in RA, some fragments with increased abundance in OA were also detected, stemming from the COL6A3 chain's 1450 and 1750 amino acid sections. Many of these degradation sites were found to map to experimentally validated MMP9 cleavage sites, as reported by MPSC and indicated by stars in Fig. 7B–D. Numerous examples of peptides within COL6A1, A2 and A3 were upregulated in RA in this dataset, experimentally validated as MMP9 cleavage sites, and scored highly in terms of protease susceptibility to MMP2 and MMP9 in this dataset. MMP9 is predominantly expressed by monocytes and neutrophils, two of the major infiltrating cell types associated with RA. Together with the association between monocytes and collagen VI degradation in tissue previously described, these proteomic data solidify a

proposed role for monocytes in the turnover of deposited collagen VI. Monocytes therefore appear to retain their migration competence over other immune cell subsets, as evidenced in Fig. 6C, through their ability to actively degrade deposited collagen VI implicating them in driving domain expansion of collagen VI "dark" zones, concomitantly releasing the detected degradation products.

Integrating transcriptomic, proteomic, degradomic and high-plex imaging in this study culminates in the discovery of perivascular collagen VI turnover as a regulator of immune cell migration in the inflamed synovium. Collagen VI has previously been reported to be overexpressed in the RA synovial sub-lining (for example, Wolf and Carsons, 1991). Indeed, our transcriptomic analysis confirms elevated COL6 mRNA in the interstitial RA synovium, and shows for the first time that levels fluctuate between phases of active disease and remission, as well as demonstrating elevated COL6 mRNA in OA synovium compared to healthy controls. Paired proteomic and degradomic analysis however reveal further complexity that dictates tissue COL6 expression; with higher myeloid-dependent, MMP2/9-mediated COL6 degradation in RA compared to OA, leading to the generation of disease-specific fragment fingerprints, a finding corroborated by the detection of COL6 degradation products in the serum of people with RA that are regulated during treatment (Thudium et al, 2024). Spatial analysis mapping matrix and immune cell distribution further demonstrates positional specificity of COL6 deposition, with COL6 accumulation in the perivascular space strongly correlated with T cells and remission-associated MerTK+ macrophages, a phenomenon which restricts cell migration in collagen VI-rich tissue niches to degradation-competent monocytes, which cell types can unlock these niches to allow immune ingress by specific proteolysis of this collagen. Together, this analysis of the synovial matrix using data spanning multiple modalities sheds new light on how synovial tissue levels of COL6 are controlled across space and time, at levels beyond transcription, to reveal the importance of dynamic matrix turnover in disease progression or resolution.

## Discussion

We have characterised the tissue composition of the RA human synovium. Matrisome-targeted GSEA identified fibroblasts, particularly of the sub-lining layer of the synovium, as the primary expressors of core ECM components, namely glycoproteins,

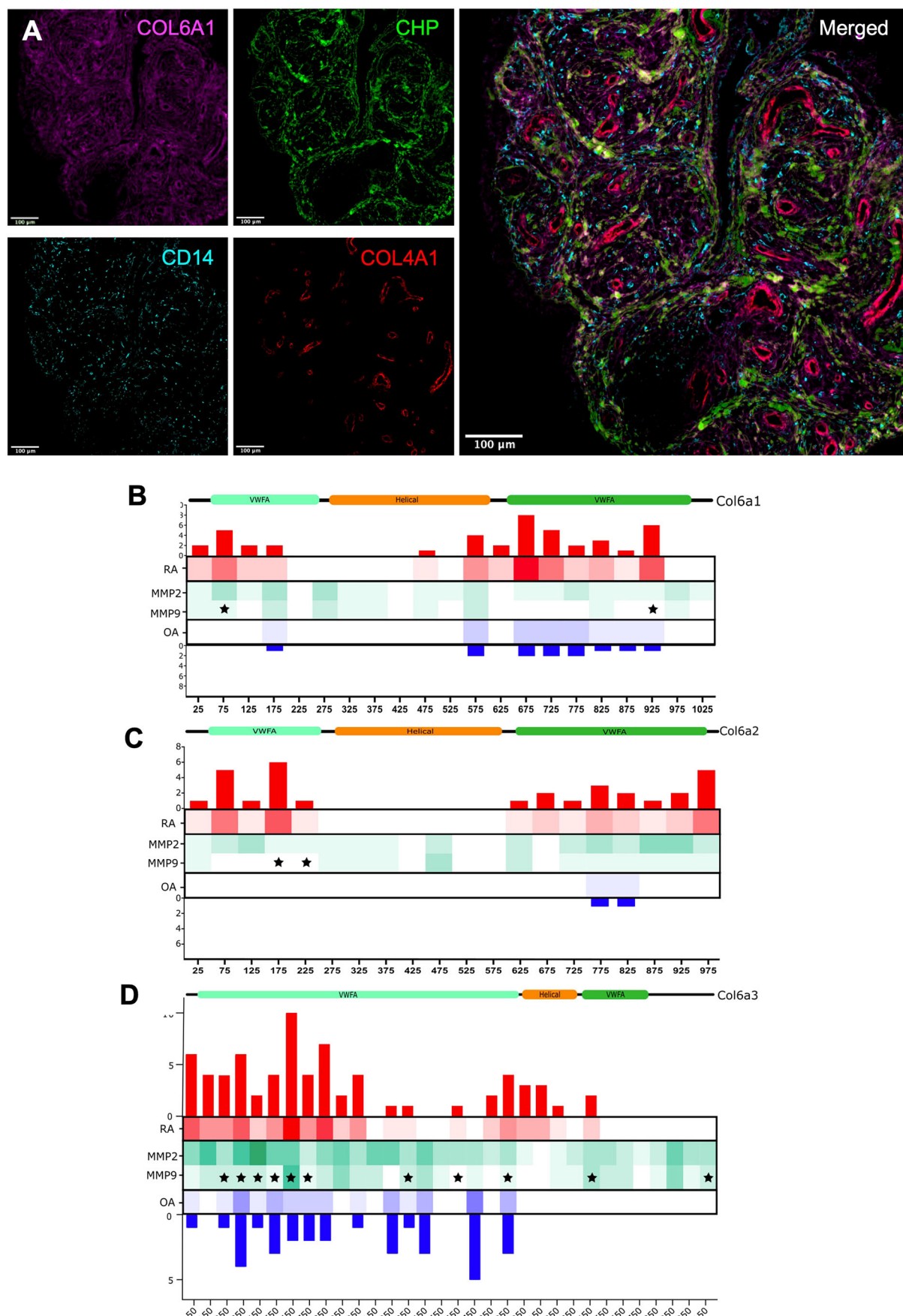

**Figure 7. Tissue monocyte infiltration is degradation-associated and the fragments from this degradation are detectable in tissue.**

(A) Immunofluorescent staining was used to characterise the spatial distribution of COL4A1, COL6A1, CHP, and CD14. QuPath was used to visualise immunofluorescent imaging of COL6A1 (magenta), CHP (green), CD14 (cyan), COL4A1 (red) in section JRP122 (representative of 3 OA and 3 RA sections). (B–D) Degraded collagen VI peptides with either increased abundance in RA (red) or OA (blue) were mapped back to 50–100 amino acid sections of the α1 (B), α2 (C), and α3 (D) chain structures. The frequency of cleavage sites within these sections was plotted in both bars and heatmaps. Corresponding heatmaps for these same sections were generated from Manchester proteome cleave (MPC) (https://www.manchesterproteome.manchester.ac.uk/#/MPSC) with a susceptibility score cut-off of 0.75 for MMP2/9 cleavage and the frequency of these were predicted cleavage sites were plotted in green. Stars indicate experimentally validated MMP9 cleavage sites. Source data are available online for this figure.

collagens, and proteoglycans. Myeloid cells were found to contribute to the expression of matrix regulators and therefore play a key role in the composition of the synovium and its dynamic remodelling. This further solidifies the incomplete role of stromal cells in defining tissue architectures, and the function of immune cell subsets, particularly myeloid cells, as matrix-mediating actors.

Clinical studies targeting fibroblasts in people with RA have not so far yielded success; for example both RG6125, an antibody that binds cadherin-11 expressed by synovial fibroblasts and ASP5094, an antibody directed against integrin α9, also highly expressed by RA synovial fibroblasts, did not show efficacy in phase 2a trials (Németh et al, 2022; Tsaltskan and Firestein, 2022; Qian et al, 2024). We have learned over the last decade that fibroblast heterogeneity reflected in distinct functional subsets may hold the key to effective manipulation of the stroma in RA, and further understanding of the diverse ways in which fibroblasts contribute to disease progression should unlock more specific and potentially more effective therapeutic strategies. Here, in characterising tissue composition, we identified clear fibroblast subset-specific matrix expression profiles which contribute to the establishment of distinct micro-architectures in the synovium. These were sufficient to cluster fibroblast subsets with striking fidelity to the original AMP1 study and point to how closely interlinked the transcriptional programs behind tissue landscape-defining profiles are with previously characterised functional profiles. Interestingly, in the highest resolution of our matrix-focused clustering, we observed the presence of the PI16 + "universal" fibroblast subset defined in both mouse and human datasets by several groups (Buechler et al, 2021; Korsunsky et al, 2022), particularly highlighted by matrix glycoprotein IGFBP5. The primary function of IGFBP5 is the inhibition of IGF1 signalling, critically dysregulated in inflammatory hyperplasic conditions such as RA, both as described in the literature and this study (Andersson et al, 2018), through direct binding to IGF1 (Kalus et al, 1998). Importantly, IGFBP5 is also strongly associated with tissue repair responses (Han et al, 2017), as well as their dysregulation in fibrosis (Nguyen et al, 2021). The presence of IGFBP5 as one of the cluster's key-defining genes may suggest a yet unexplored role for this "universal" subset in injury repair responses and ensuring tissue-level returns to homeostasis. As exemplified by this, we believe an interesting knowledge gap to be the conservation of these newly identified fibroblast matrix signatures across tissues with distinct function-driven designs and geographies.

The cross-disease tissue remission signature identified in this study stems from such an approach. FS is characterised by shoulder pain and immobility due to chronic inflammation and fibrotic-like tissue remodelling of the shoulder capsule, features bearing a striking resemblance to the inflammatory tissue remodelling of the synovium in RA. FS, in stark contrast to RA, is self-limiting and resolves in the majority of patients in the natural course of the condition (Hand et al, 2008). Because of the similarities in homeostatic tissue composition and inflammatory natures of these conditions, we looked for overlapping resolution signatures of DEGs between RA in remission and FS, to define the matrisomal features of a tissue in remission. The key pathways to come out of this analysis were those of IGF-1 and TGF-β signalling modulation, particularly through expression of LTBP4, HTRA4, CILP, IGFBP5, and IGFBP6. Extensive literature exists surrounding the role of both signalling pathways in RA. IGF-1 signalling is heavily implicated in RA pathology, with early in vivo studies suggesting mild positive effects from its inhibition on IL-6 levels and fibroblast proliferation (Erlandsson et al, 2017; Zhang et al, 2019). The availability of drugs such as Metformin and Teprotumumab, with described indirect and direct effects on IGF-1 inhibition, respectively, should enable the effective exploration of this pathway as a therapeutic target (Xie et al, 2014; Douglas Raymond S et al, 2020).

In line with our data, elevated TGF-β signalling is also widely observed in people with RA and in murine disease models (reviewed in Gonzalo-Gil and Galindo-Izquierdo, 2014). This appears paradoxical at first glance; TGF-β displays well-documented immune suppressive roles, including promotion of Treg expansion, inhibition of effector T-cell function, and switching of macrophages and neutrophils towards an anti-inflammatory phenotype (Batlle and Massagué, 2019). However, TGF-β is also extremely pleiotropic and can drive pro-inflammatory responses, for example, stimulating secretion of cytokines such as TNF-α and IL-1, activating proteases including MMPs, and acting as a chemoattractant for neutrophils (Gorelik and Flavell, 2001; Gorelik et al, 2002). Moreover, it participates in many diverse aspects of RA pathology, including stimulation of angiogenesis, and sustained fibroblast hyperplasia and activation, which together make it difficult to precisely dissect the net effect of elevated activity in RA (Bira et al, 2005; Sakuma et al, 2007; Bhamidipati et al, 2025). Indeed, attempts to inhibit TGF-β signalling have yielded mixed results in murine models of arthritis with some studies showing amelioration of disease and others exacerbation (Sakuma et al, 2007; Aarts et al, 2022). The tissue microenvironment provides key context that defines the impact of elevated TGF-β on cell behaviour and immune status. This is driven at least in part by coordinated changes in the overall cytokine cocktail of the RA synovium. For example, elevated IL-6 and Il-23 in RA may direct TGF-β-mediated Th17 cell generation at the expense of Treg expansion. Indeed, TGF blockade in vivo reduces pathogenic, bone-destroying Th17 cell abundance and increases Treg numbers (Aarts et al, 2022). The role of TGF-β in RA is therefore complex and exquisitely context-specific, and warrants further investigation,

taking into account disease stage or activity and cell-type-specific responses.

Through in-depth characterisation of the synovium ECM, we found dysregulated expression of *COL6A1*, *COL6A2* and *COL6A3*, and known collagen VI regulators (*MMP2, MMP9, TIMP3, TIMP4*) to be a key feature of fibroblasts in active RA. Elevated transcription of COL6 genes in RA is consistent with elevated TGFβ signalling in RA; this growth factor is the best studied inducer of this collagen to date. Direct treatment of cells with TGFβ induces COL6 mRNA expression, a process mediated by binding of transcription factors Sp1 and SREBP to sites proximal to the transcriptional start site in the COL6 promoter (Piccolo et al, 1995; Ferrari et al, 2004) and involving histone acetylation by CREBBP and EP300 (Williams et al, 2020). However, lower COL6 expression in CTAP:F and higher expression in CTAP:M is at odds with the known association of this collagen with fibrosis in a number of different organs. Studies showing that COL6 expression can be induced by factors produced by alternatively activated macrophages, including TGFβ, but also other cytokines such as IL4 and IL10 (Schnoor et al, 2008) may explain this observation. These data mirror the correlation between COL6 expression in the synovium and MERTK+ macrophages and may help to explain the temporal regulation of COL6 that we observe in resolving experimental murine arthritis. In this model, expression is elevated during the end stages of inflammation, at the start of disease resolution, which may reflect chronically elevated TGFβ signalling in active RA causing prolonged COL6 transcription. Of note, TNF-α activation of intestinal myofibroblasts caused the production of an ECM with increased COL6 protein detected by mass spectrometry, whilst TGFβ-treated cells produced an ECM containing lower levels of COL6 protein (Lin et al, 2022) further highlighting complexities between gene expression and protein levels of this collagen.

Despite the relative overexpression of collagen VI genes in RA, we hypothesised that the accompanying overexpression of *MMP2* and *MMP9* in active disease, contrasting with the overexpression of their inhibitor TIMP3 in remission, would contribute to net protein deposition in the remission phases of disease. Staining of both human synovial biopsies and mouse AIA model sections corroborated this, with increased deposition of collagen VI in remission-associated states. Spatial analysis of collagen VI tissue distribution unveiled areas of weaker COL6A1 staining, termed "COL6 dark" zones, in which CD45+ immune cells were retained. Interestingly, such structures were found in synovial inflammation, OA synovium, IBD, and head and neck cancer. This observation is reminiscent of the observation of Martinez and colleagues, in which reduced collagen VI staining intensity and fibre integrity accommodate lymph node expansion in immune challenge models (Martinez et al, 2019). Based on the location of COL6 dark zones, which arise at the vasculature basement membrane, often extending deep into the synovial (and intestinal and tumour) tissue, we aimed to model in vitro immune cell movement within tissues using FN as a migration-permissive matrix molecule that is ubiquitously expressed in the stroma of all tissues including the synovium. Both monocytic and T-cell lines readily migrated on FN but were unable to migrate on collagen VI alone; cell movement was also blocked on migration-competent surfaces (FN) coated with collagen VI. This is consistent with studies showing that

T-cell migration on collagen VI substates is reduced compared to migration on either FN or collagen I, which is caused by disrupted fibrillar actin formation required to maintain efficient motility and traction force on collagen VI that does not occur in cells interacting with FN or collagen I (Pruitt et al, 2023). These data are also consistent with data showing that collagen VI, but not collagen I or IV, significantly blocks monocyte migration in a transendothelial migration assay across activated HUVECS. In line with our data, collagen VI prevented T cell migration across the endothelial cell barrier, however, interestingly, in this assay collagen I also inhibited T cell movement across a HUVEC substrate. These data suggest that lymphocytic migration occurs via different mechanisms depending on the underlying substrate, and which have different susceptibility to collagen modulation (Nisa et al, 2024). Together, these data show that collagen VI alone does not support effective monocyte or T cell migration, and that this collagen also restricts migration of both cell types on permissive substrates, regardless of the nature of the underlying substrate. These data also demonstrate that, whilst collagen I does not impact monocyte transendothelial migration, it does prevent T cells moving across the vascular barrier. Together, these data highlight a series of sequential checkpoints that monitor cell ingress into tissues at different stages.

The deposition of collagen VI has been reported as a feature of tissue remodelling and repair, from healthy scar tissue formation (Jimenez-Mallebrera et al, 2006), to pathogenic chronic remodelling of fibrosis (Williams et al, 2020), being essential for neuromuscular regeneration (Urciuolo et al, 2013; Chen et al, 2015; Takenaka-Ninagawa et al, 2021). This work extends the scope of the collagen VI literature to chronic inflammatory tissue remission, while also shedding light on a novel role for collagen VI. Indeed, we demonstrate that collagen VI is deposited in the perivascular niche as part of the tissue repair process to restore tissue integrity, but also to lock inflammatory immune cells out of the deeper tissue by preventing their migration beyond the point of entry.

The implications of our findings highlight the importance of tissue microenvironment composition in the spatial distribution or restriction of immune cells in disease. Reverse-engineering these restrictive properties may provide novel strategies for putting "brakes" on specific immune subsets in diseases characterised by dysregulated adaptive immune infiltration, such as RA.

Understanding that net collagen VI deposition is the result of a dynamic balance of proteases and their inhibitors, we looked to notable protease-expressing cell subsets, particularly monocytes and neutrophils, as potential key players in the formation and maintenance of the "COL6 dark" zones. Interestingly, CHP staining of the "dark" zones revealed active degradation sites around the outermost borders of the "dark" zones, overlapping with the outermost layers of collagen VI-rich areas. These ring-like borders were found to be predominantly lined with monocytes. In addition, monocytes were found to be the only cell subsets able to enter the deeper tissue. This tissue ingress was also stained by CHP, suggesting a degradation-associated migration process. Together these data put monocytes in the spotlight as the infiltrating cells likely to be responsible for the establishment and expansion of the "COL6 dark" zones, which may be acting as adaptive immune

outposts in peripheral tissue, and which could explain the higher rate of collagen turnover in myeloid-rich RA endotypes as well as the lower rate of collagen VI turnover in comparatively myeloid-poor OA synovium. These results imply that potential benefits may be derived from targeting the proteases necessary for "COL6 dark" zone expansion and inflammatory monocyte and neutrophil tissue ingress. This is corroborated by mouse studies of inflammatory arthritis showing reduced pathology in *Mmp9* KO mice (Itoh et al, 2002), as well as a Phase 1b clinical trial of the anti-MMP9 antibody Andecaliximab with modest impact on efficacy (Gossage et al, 2018). Another approach could be to harness endogenous inhibition systems to restore a favourable balance between proteases and their inhibitors and prevent uncontrolled chronic cellular infiltration.

The described active collagen VI degradation process suggests the release of collagen VI fragments into the perivascular "COL6 dark" zones, and possibly the bloodstream itself. In line with this, collagen VI fragments were readily detected by mass spectrometry in synovial tissue. This observation is strengthened by work from Nordic Bioscience, identifying collagen VI degradation fragment C6Mα3 as a biomarker of disease severity in IBD, and collagen VI deposition biomarker PRO-C6 as associated with disease remission (Lindholm et al, 2021). More recently, the company also described changes in serum levels of collagen VI degradation fragment C6M in the first 16 weeks after treatment with Tocilizumab as a potential indicator of treatment efficacy (Thudium et al, 2024). In addition to their interesting applications as biomarkers, these degradation products may also serve specific functions as matrikines, with implications for inflammatory disease pathology. Indeed, collagen fragments have been reported to have numerous effects on cellular phenotypes and activities, ranging from pro-fibrotic and pro-tumorigenic effects for endotrophin (Kim et al, 2020; Bu et al, 2019), to anti-angiogenic and anti-tumorigenic effects for endostatin (O'Reilly et al, 1997). Such specific immunomodulating functions would be crucial to explore in future studies of collagen VI degradation products.

In summary, our data suggest a model wherein type VI collagen expression by fibroblasts is induced during the resolution stages of synovial inflammation as a result of high tissue levels of TGFβ and/or of remission-associated macrophages, preventing further immune cell ingress. Following additional insult or re-challenge, MMP2/9-licenced infiltrating monocytes degrade peri-vascular collagen VI, clearing a path for subsequent cellular entry and releasing collagen VI fragments. This effect is specific to collagen VI but is not limited to the RA synovium. Induction of COL6 gene expression in OA also creates immune-restrictive perivascular niches, but here protein is less readily degraded than in RA, likely due to more modest myeloid infiltration in this disease. These niches also arise in inflammatory diseases in other tissues and in immune-infiltrated tumours, suggesting this means of controlling immune infiltration is a universal response of fibroblasts.

There are several limitations to this study. The study focused on collagen VI and its processors from among a broader list of prioritised DEGs. Other genes and their relevant proteins likely play a role in the pathological remodelling of the synovium in inflammatory arthritis. Future studies could build on this work by creating a wholistic model of synovial matrix dysregulation rather than focusing on individual targets.

# Methods

**Reagents and tools table**

| Reagent/resource | Reference or source | Identifier or catalogue number |
|---|---|---|
| **Experimental models** | | |
| THP-1 cells (*H. sapiens*) | ATCC | TIB-202 |
| Jurkat E6-1 cells (*H. sapiens*) | ATCC | TIB-152 |
| C57BL/6J (*M. musculus*) | Kennedy Institute of Rheumatology animal facility | N/A |
| **Antibodies** | | |
| Rabbit anti-human MMP2 | Atlas Antibodies | HPA001939 |
| Collagen Hybridising Peptide | Cell Systems | 5276-60UG |
| Mouse anti-human CD66b | Biolegend | 305110 |
| Rabbit anti-human COL3A1 | Abcam | ab237238 |
| Mouse anti-human MMP9 | Santa Cruz | sc-21733 |
| Rabbit anti-human CD14 | Abcam | ab133335 |
| Rabbit anti-human COL1A1 | Cell Signalling Technologies | 28368S |
| Mouse anti-human COL12A1 | Santa Cruz | sc-166020 |
| Rabbit anti-human COL6A1 | Abcam | ab200430 |
| Rabbit anti-human CD3E | Abcam | ab245731 |
| Rabbit anti-human Fibronectin | Abcam | ab275110 |
| Rabbit anti-human CD20 | Abcam | ab198943 |
| Rabbit anti-human MERTK | Abcam | ab271851 |
| Rabbit anti-human CD4 | Abcam | ab280849 |
| Mouse anti-human COL4A1 | Invitrogen | 51-9871-82 |

| Reagent/resource | Reference or source | Identifier or catalogue number |
| --- | --- | --- |
| Rabbit anti-human CD45 | Abcam | ab282747 |
| Rabbit anti-human CD68 | Abcam | ab280860 |
| Mouse anti-human CD8 | Biolegend | 372902 |
| Rabbit anti-human hIgG | Abcam | ab226069 |
| Rabbit anti-human CD11c | Abcam | ab279329 |
| Rabbit anti-human MZBI | Novus Biologicals | NBP2-90320 |
| Rabbit anti-human MMP2 | Abcam | ab237473 |
| Rabbit anti-human CD11b | Abcam | ab133357 |
| Rabbit anti-human LYVE1 | Abcam | ab219556 |
| Rabbit anti-human COL1A1 | Abcam | ab138492 |
| Rabbit anti-human MFAP5 | Abcam | ab240367 |
| Rabbit anti-human MMP1 | Abcam | ab196905 |
| Rat anti-human PDPN | Invitrogen | 14-9381-82 |
| Rabbit anti-human IGFBP5 | Invitrogen | PA5-37369 |
| Goat anti-human IGFBP7 | R&D Systems | AF1334 |
| Rabbit anti-human NaKATPase | Abcam | ab197713 |
| Mouse anti-human RS6 | Santa Cruz | sc-74459 |
| Rabbit anti-human Cadherin | Abcam | ab195203 |
| Rabbit anti-human Vimentin | Abcam | ab185030 |
| Rabbit anti-human CD19 | Abcam | ab196468 |
| Rabbit anti-human MERTK | Abcam | ab52968 |
| Rabbit anti-human CD68 | Abcam | ab277276 |
| Rabbit anti-human CD34 | Abcam | ab81289 |
| Rabbit anti-human CD90 | Abcam | ab181885 |
| Rat anti-mouse CD3 | Biolegend | 100228 |
| Rat anti-mouse CD68 | Biolegend | 137012 |
| Rabbit anti-mouse CD177 | R&D Systems | MAB8186 |
| Rat anti-mouse CD31 | Biolegend | 102520 |
| **Chemicals, enzymes and other reagents** | | |
| Ammonium-Chloride-Potassium (ACK) Lysing Buffer | Gibco | A1049201 |
| Bovine Serum Albumin (BSA) | Sigma-Aldrich | A7906 |
| Carbo-free blocking solution 10x | 2B Scientific | SP-5040-125 |
| CellPath Optimal Cutting Temperature (OCT) Embedding Matrix | CellPath | 15212776 |
| Stock antigen unmasking solution | Vector Laboratories | H-3300 |
| 4′,6-diamidino-2-phenylindole (DAPI) Stock Solution | ThermoFisher Scientific | D3571 |
| Donkey Serum | Sigma-Aldrich | D9663 |
| Ethanol | Honeywell | 32221-2.5 L |
| Foetal Bovine Serum (FBS) | Gibco | 10500-064 |
| Fc Receptor Blocking Reagent | Miltenyi | 130-059-901 |
| FluorSave Reagent | Merck | 345789-20 ML |
| Human TruStain FcX™ (Fc Receptor Blocking Solution) | Biolegend | 422302 |
| Hydrogen Peroxide | Sigma-Aldrich | 216763 |
| Kimberly-ClarkTM KimtechTM Science Precision Wipes (Kimwipes) | Fisher Scientific | 11768188 |
| Leica SurgiPath Coverglass 24 ×50 mm coverslips (Size 1) | Leica | 3800145 G |

| Reagent/resource | Reference or source | Identifier or catalogue number |
|---|---|---|
| Pap pen | ThermoFisher | 8899 |
| 10% Paraformaldehyde (PFA) | ThermoFisher | 47317.9 L |
| Penicillin Streptomycin | Gibco | 15140-122 |
| Phosphate Buffered Saline (PBS) 10X | VWR | 437117 K |
| RMPI 1640 | Gibco | 21875-034 |
| Stock Antigen Unmasking Solution | Vector Labs | H-3300-250 |
| Super Frost Plus Glass Slides | Fisher Scientific | 10149870 |
| SyTox Blue dye | ThermoFisher | S11348 |
| Tris base | ThermoFisher | T1503 |
| Triton-X-100 | Sigma-Aldrich | T9284 |
| Trizma® Base | Sigma-Aldrich | T6066 |
| Tween 20 | VWR | 663684B |
| Xylene | Sigma-Aldrich | 534056 |
| β-mercaptoethanol | Sigma-Aldrich | M6250 |
| Human native Fibronectin | Abcam | ab80021 |
| Human native collagen VI | Abcam | ab7538 |
| **Software** | | |
| GraphPad PRISM 10.2.2 | https://www.graphpad.com | N/A |
| R version 4.0.3 (2020-10-10) | https://www.r-project.org/ | N/A |
| QuPath 0.5.0 | https://qupath.github.io/ | N/A |
| Proteome Discoverer 3.1 | https://www.thermofisher.com/uk/en/home/industrial/mass-spectrometry/liquid-chromatography-mass-spectrometry-lc-ms/lc-ms-software/multi-omics-data-analysis/proteome-discoverer-software.html?erpType=Global_E1 | N/A |
| Perseus 2.0.11 | https://maxquant.net/perseus/ | N/A |
| Graphia 4.2 | https://graphia.app/ | N/A |
| **Other** | | |
| Biocare Decloaker | Biocare Medical | https://biocare.net/products/instrumentation/declo aking-chamber-nxgen/ |
| Cell DIVE multiplexed imaging platform | General Electric | https://www.ge.com/research/project/multiplexed-tissue-imaging-platform |
| Incucyte® SX5 | Sartorius | https://www.sartorius.com/en/products/live-cell-imaging-analysis/live-cell-analysis-instruments/sx5-live-cell-analysis-instrument |
| Leica CM3050 cryostat | Leica | https://www.leicabiosystems.com/en-gb/histology-equipment/cryostats/leica-cm3050-s/ |
| Manual Rotary Microtome HistoCore BIOCUT | Leica | https://www.leicabiosystems.com/en-gb/histology-%20equipment/microtomes/hi stocore-biocut/ |
| ZEISS LSM 980 Confocal | ZEISS | https://www.zeiss.com/microscopy/en/products/light-microscopes/confocal-microscopes/lsm-980-with-airyscan-2.html?utm_source=google&utm_medium=cpc&utm_campaign=C-00011305&gad_source=1 |

## Bioinformatic analysis

Unless otherwise stated, all bioinformatic analyses were performed using R version 4.0.3 (2020-10-10), in RStudio version 1.3.1093 for MacOS. Plots, unless otherwise stated, were generated using ggplot2 version 3.3.5.

## Datasets

Single-cell RNA sequencing datasets used and re-analysed in this study are listed below (Table 1).

### AUCell gene set enrichment analysis

Mapping of the RA synovial matrix was performed both on the Alivernini datasets using the *Seurat* and *AUCell* packages (Aibar et al, 2017; Hao et al, 2021). The major matrix and matrix subcategory-expressing cells were identified using the AUCell package following the documentation vignette (available through *vignette("AUCell")* in R). Notable changes made to the script laid out in the vignette were the use of the matrisome subcategories defined by the Matrisome Project as the gene sets of interest (Naba et al, 2016), and the increase of the threshold of cells used for gene set enrichment calculations from 5 to 10%. The gene set

enrichment scores for each cell and each matrisome subcategory, or AUC scores, were then used to generate the boxplots and feature plots, using the *ggplot2* package.

### Pseudotime analysis

Pseudotime analysis was performed on the AMP1 dataset using the *slingshot* package (Street et al, 2018). Fibroblasts were computationally subsetted from the rest of the dataset's cells, and the lining layer SC-F4 were selected as the starting cluster for trajectory analysis based on their discrete physiological location relative to sub-lining SC-F1/2/3. Trajectories were plotted in PCA space. Temporally expressed genes and matrix genes were then identified by fitting a Generalised Additive Model, with a LOESS term for Pseudotime. These were plotted using the *pheatmap* package (Kolde, 2019).

### gsfisher gene set enrichment analysis

Gene Set Enrichment Analysis was performed using the *gsfisher* package, following the documentation vignette available in the package's GitHub repository (https://github.com/sansomlab/gsfisher).

### Alluvial label transfer

Comparisons between the de novo-clustered and matrisome-clustered Alivernini fibroblasts were visualised using alluvial plots from the *ggalluvial* package (Brunson, 2020). Metadata columns containing the annotated clusters of each cell using each method were taken from the analysed *Seurat* object, enumerated, and wrangled into a data frame that could be taken by the package as input.

### De novo fibroblast clustering

To enable direct comparisons across datasets, Alivernini dataset cells were integrated and clustered de novo using the current gold standard method for scRNAseq data integration, Harmony (Korsunsky et al, 2019). Data from remission and active RA samples from the Alivernini dataset were integrated using Harmony using the samples of origin as the covariates. A first round of clustering was then performed in *Seurat*, with a resolution of 0.3, to identify and subset stromal cells based on the expression of PTPRC- and PECAM1+, FAP+ or PDPN+. All clustering steps were performed using the *clustree* package to select the ideal resolution to optimise cluster robustness (Zappia and Oshlack, 2018). A second round of clustering was done, with a resolution of 0.3 again, to identify fibroblasts based on the expression PECAM1- and PDPN+, FAP+ or THY1+. With the fibroblasts computationally sorted, a third round of clustering was performed with a resolution of 0.5, which revealed high levels of condition-specific clusters, hence unlikely to be biologically relevant. To account for these effects, likely due to the large number of cells present in the initial integration, the cell identification codes for all the sorted fibroblasts were extracted and used to subset only fibroblasts from the original non-integrated dataset, and a new round of Harmony was run in these cells only. Fibroblast-specific integration was considerably improved as a result, and a final round of integration with 0.4 resolution revealed AMP-like clusters, used as input for the Differential expression analysis (DEA) pipeline.

### Matrisome-focused clustering

De novo clustered fibroblasts from the Alivernini dataset were also subjected to matrix-focused clustering. The matrisome gene list was first manually curated to create a new list excluding cytokines and chemokines from the Secreted Factors category of the matrisome (Fig. S1). This curated list was then used as the *features* input for the *RunPCA()* function of the *Seurat* clustering pipeline, limiting the PCA input to curated matrisome genes and thereby limiting the downstream graph-based clustering to be defined by differences in matrisome gene expression. The basic level of matrix clustering presented in this study was performed with a resolution of 0.3, while the deepest level presented here was performed with a resolution of 0.5.

### Differential expression analysis

The Seurat pipeline described in the relevant vignette (https://satijalab.org/seurat/articles/pbmc3k_tutorial.html) was used to identify cluster-defining markers between fibroblast subsets in each dataset. Specifically, the *FindAllMarkers()* function was used, testing only for genes detected in at least 25% of cells within the cell subset and with an average log-fold change of at least 25% between two cell subsets, and using the default Wilcoxon Rank Sum test to identify differentially expressed genes between clusters. A more stringent differential expression analysis (DEA) pipeline was also set up and applied to the AMP and Alivernini datasets to compare OA and RA, and remission and active RA samples, respectively. The pipeline comprises of two established DEA methods, edgeR and MAST (Robinson et al, 2010; Finak et al, 2015), which both form part of the best practice in single-cell analysis at the time of the analysis (Luecken and Theis, 2019). They are used here in parallel to provide an additional level of coverage of and confidence in potentially differentially expressed genes (DEGs). The stringent bulk RNAseq method edgeR was run following the relevant documentation vignette (available through *edgeRUsersGuide()* in R) on raw counts data sum-pseudobulked by cell cluster and sample and with the *P* values taken from the output of the *glmLRT()* function of the *edgeR* package. MAST was run directly on the log-normalised data using the *Seurat FindMarkers()* function, and *P* values were adjusted using the *p.adjust()* function using the Benjamini–Hochberg adjustment approach (Benjamini and Hochberg, 1995).

### Target prioritisation

To sieve through the large number of DEGs identified by both methods, a target prioritisation scheme was set up combining three different metrics. The first metric is the overlap between the DEA methods. The second metric is the conservation of the DEG expression pattern in the Wei dataset analysed using MAST, comparing control mice and Serum Transfer-Induced Arthritis (STIA) mice (Wei et al, 2020; Data ref: Wei et al, 2020). The third and final metric is a differential expression score calculated as:

$$| - log10(pval).log2(FC)|$$

The latter component avoids the common analytical downfall of defining genes as "statistically significant", which is heavily caveated, particularly when working with scRNAseq data. It relies instead on placing genes on a spectrum of differential expression.

### Independent data set analysis

To examine expression of collagens in healthy individuals and in a larger disease cohort, we utilised published single-cell datasets (healthy knee, n = 13) (Data ref: Faust et al, 2024), OA (n = 10) and RA (n = 88) (Data ref: Thomas et al, 2024). The RA dataset is a

**Table 1.   Publicly available datasets used for bioinformatic analyses.**

| Species | Type | Tissue/cells | Disease/model | Source | Accession no. | In-text name |
|---|---|---|---|---|---|---|
| Human | scRNAseq | Sorted T-cells/B cells/Macrophages/Fibroblasts from synovial biopsy | OA and RA patients | Data ref: Zhang et al, 2019 | SDY998 | AMP1 |
| Human | scRNAseq | Unsorted cells from synovial biopsy | Active RA and RA in remission patients | Data ref: Alivernini et al, 2020 | E-MTAB-8322 | Alivernini (part of combined dataset) |
| Mouse | scRNAseq | Unsorted synovial cells from synovial dissections | STIA | Data ref: Wei et al, 2020 | GSE145286 | Wei |
| Human | scRNAseq | Unsorted cells from the shoulder capsule | FS and comparator patients | Data ref: Ng et al, 2024 | ERP143358 | FS |
| Human | scRNAseq | Sorted live cells from synovial samples | Healthy synovium | Data ref: Faust et al, 2024 | GSE233500 | Faust (part of combined dataset) |
| Human | scRNAseq | Unsorted cells from synovial samples | OA and RA patients | Data ref: Zhang et al, 2023 | https://doi.org/10.7303/syn52297840 | AMP2 (part of combined dataset) |
| Human | scRNAseq | Unsorted cells from synovial samples | RA patients | Data ref: Thomas et al, 2024 | https://doi.org/10.5281/zenodo.13768607 | TAURUS-RA (part of combined dataset) |

scRNAseq single-cell RNA sequencing, OA osteoarthritis, RA rheumatoid arthritis, STIA serum transfer-induced arthritis, FS frozen shoulder.

combination of data from the TAURUS, AMP2 (Data ref: Zhang et al, 2023) and Alivernini (Data ref: Alivernini et al, 2020) cohorts. For analysis across all 3 states, patients in active remission were removed, owing to previously validated reduced expression of collagens. Dot plots of log-normalised collagen and collagen-binding partner expression were plotted for previously annotated cell types. To test differences in collagens between OA and RA, in the larger cohort, again we utilised the combined dataset (Data ref: Thomas et al, 2024) with patients undergoing remission removed, leaving $n = 10$ OA patients and $n = 88$ RA patients. Previously annotated fibroblast clusters were subsetted and data was pseudo-bulked by patient and cluster using the Seurat (version 5.2.1) function AggregateExpression. Comparison of fibroblast collagen expression, between OA and RA patients, revealed variable expression across RA patients. To decipher factors driving this variation, we limited the pseudobulked dataset to RA patients only and ran a linear regression model, testing the effects of joint site and CTAP on expression of collagens across each fibroblast cluster. Both site and CTAP significantly influenced expression of collagens, compared to the reference, across multiple clusters. Consequently, to formally test expression of collagens between OA and the different RA CTAPs, we again filtered the dataset to samples acquired from knee joints only, and in which CTAP status was available (AMP2 cohort, $n = 7$ OA & $n = 29$ RA patients). A generalised linear model framework (EdgeR, version 4.0.16) was used on pseudobulked data (aggregated on patient and cluster) to formally test differential expression of collagens between OA and each RA CTAP. $P$ values were adjusted for multiple testing and an adjusted $P$ value $< 0.05$ were considered significant.

### Type VI collagen interactome analysis

MatrixDB (https://matrixdb.univ-lyon1.fr/) was queried for protein-interacting partners of the α1 (P12109), α2 (P12110), and α3 (P12111) chains of type VI collagen. The output of this search was three lists of 91, 280, and 290 predicted and experimentally validated interacting partner proteins. These were combined into a list of 457 unique collagen VI interactors. This list of collagen VI interactors was used as input for a STRING (https://string-db.org/, v12.0) network and pathway analysis. The resulting 20 most enriched Biological Processes GO terms, KEGG pathways, and Reactome pathways were selected for plotting. Protein binders were filtered for high confidence, physical interactions and this network was exported. Proteins with one or more interactions were included as nodes and edges were weighted by their STRING interaction score. Network modularity was investigated as previously described (Llewellyn et al, 2023), to examine proteins that clustered into independent, highly-interconnected modules.

### Proteomic analysis

Raw data was accessed from PRIDE, project PXD027703 (Ren et al, 2021; Data ref: Ren et al, 2021); PXD020397 (Buschhart et al, 2020; Data ref: Buschhart et al, 2020), and PXD016620 (Lee et al, 2020; Data ref: Lee et al, 2020). The first dataset was re-analysed using Proteome Discoverer (version 3.1) using the TMTe 6plex quantification method, the second and third datasets were re-analysed using Fragpipe (v23). Spectral counts were used to limit technical differences between datasets where appropriate. All were searched against the UniProt/SwissProt human database (2024)

with added decoys and known contaminants which contained a total of 40,936 sequences (50% decoys) with a maximum false discovery rate (FDR) of 1%. Fixed modifications included in the searches were carbamidomethyl addition to cysteine ($+57.021$ Da) and TMT 6-plex modifications at peptide N-termini and lysine residues ($+229.163$ Da). Dynamic modifications of oxidation of methionine ($+15.995$ Da), hydroxylation of proline ($+15.995$ Da), deamination of asparagine/glutamine ($+0.984$) and phosphorylation of serine/threonine/tyrosine ($+79.966$) were also included. 2 missed cleavages were allowed and both tryptic and semi-tryptic searches were run. Peptides were only included if they met the threshold 1% FDR and known contaminants were removed. Data was exported and onward analysis was conducted in Perseus (v2.0.11), R (v4.3.2) and Graphia (v4.2). Data were normalised using median subtraction. Semi-tryptic to tryptic ratios were calculated from average abundances (for tmt labelled samples) or spectral counts of the relevant peptides for the proteins of interest, following removal of N-terminal peptides. Protein sequence mapping was done using normalised semi-tryptic peptide abundance. Network analysis was performed using Graphia (Freeman et al, 2022). Proteins are modelled as nodes and the edges between them are based on correlation in expression. Edges were only included if their Pearson Correlation Value was over 0.85. Within this, the data was organised based on Louvain clustering with a granulation of 0.5. MMP2/9 susceptibility and cleavage sites were taken from Manchester Protease Susceptibility Calculator.

### Dataset background
[PXD027703]: Synovial tissue from RA or OA patients undergoing surgical joint replacement, diagnosis of the patients was accorded to the criteria of the American College of Rheumatology (ACR) and European League Against Rheumatism (EULAR) in 2010. 10 RA or 12 OA patients were split into 3 groups and each group was pooled together to produce 3 MS samples for each condition. [PXD044963]: Synovial biopsies from RA patients part of the Pathobiology of Early Arthritis Cohort (PEAC). Seven pre-treatment patient samples were collected, four displaying myeloid and three lymphoid RA pathotypes. [PXD016620]: OA synovial fluid was collected from ten osteoarthritic patients selected using the American College of Rheumatology criteria from Keimyung University Dongan Hospital in South Korea.

### Cell subtype distribution in collagen VI-rich and poor regions
The tissue sections stained for the validation panel of markers and imaged on the CellDIVE system were run through the SCIMAP Python toolkit for in-depth analysis of extracellular collagen VI fibres and cell phenotyping by Shihong Wu (Nirmal and Sorger, 2024). Following the pipeline, cells and collagen VI fibres were segmented and cellular subtypes were assigned based on the decision matrix laid out in Appendix Table S1. Local fibre density scores were calculated for each individual fibre through nearest neighbour analysis, and each fibre was assigned to COL6A1-rich or COL6A1-poor areas. This assignment was based on a cut-off value set at the median local fibre density of each section, which was reproducibly found to be 0.01. Fibres were then clustered, and the outermost fibres were selected to label the boundaries of the COL6A1-rich and poor regions. The coordinates of these fibres were then joined up into a polygon, which was expanded further by 50 μm to define the regions of interest. Cell subtype frequencies were then measured in each of the regions and the frequencies were exported into R, where a Fisher's exact test was run.

## Immunofluorescence

### *CellDIVE multiplexed immunofluorescence—human tissue*
Multiplexed immunofluorescence (mIF) imaging was performed on the CellDIVE (Leica Microsystems). Synovial tissue samples from OA and RA patients were obtained from the University of Oxford (Prof. Sarah Snelling) and Birmingham (Mr. David Gardner) (Table 2). Samples were obtained with written, informed consent and approval from the West Midlands Black Country Research Ethics committee (12/WM/0258) and the Oxford Musculoskeletal Biobank (09/H0606/11 and 19/SC/0134) in compliance with the Declaration of Helsinki. After tissue collection, biopsies were fixed in 4% PFA for 6–18 h and formalin-fixed paraffin-embedded (FFPE) block preparation was performed according to the histology unit guidelines specific to each institute. FPPE blocks were sectioned on a microtome at 5 μm thickness and deposited onto SuperFrost Plus™ Glass slides. Immediately after sectioning, slides were baked in a laboratory oven at 60 °C overnight to promote tissue adhesion. This step is crucial as ensuring the adhesion of tissue sections to slides enables the extensive rounds of staining. After baking, the slides could be kept at room temperature for storage.

From here, slides were baked once more overnight at 60 °C. The slides were removed from the oven, and deparaffinised through plunging into xylene solutions twice for 5 min, followed by decreasing concentrations of ethanol (100%, 95%, 70%, 50%) twice for 5 min each. Slides were then plunged into 0.3% Triton-X-100 in 1xPBS for 10 min for tissue permeabilization. Slides were then washed twice for 5 min in PBS.

Tissue antigen retrieval was performed in a NxGen Decloaking Chamber, set up to heat up to 110 °C and maintain this temperature for 4 min. When the chamber temperature reached 70 °C, slides were placed in a pH = 6 citrate-based solution (ARS1) for 20 min, with the temperature rising to 110 °C and pressure building to 6 psi during that time. After 20 min, pressure was released and slides were moved to a pH = 9 Tris-based solution (ARS2) for 20 min. After 20 min, slides were allowed to cool down on the bench while remaining in the Tris-based solution (ARS2). Once cool, slides were washed 4 times for 5 min in $1 \times$ PBS. Finally, slides were left in a solution of 10% donkey serum, 3% BSA in $1 \times$ PBS overnight at 4 °C to block unspecific antibody binding sites. Slides were then plunged into a DAPI solution for 15 min, and washed for 5 min in $1 \times$ PBS. Finally, 75 μL of mounting media (50% glycerol and 4% propyl gallate in $1 \times$ PBS) was added to each slide, before being coverslipped and stored at 4 °C.

Slides were then ready to be imaged. Background imaging of the slides on the CellDIVE was performed within two days of tissue processing. After background imaging, slides were decoverslipped by being angled face down in a container filled with $1 \times$ PBS for 15 min to 1 h. Slides were then carefully dried with Kimwipes and placed into a staining dish. Slides were Fragment crystalisable (Fc)-blocked using Fc Receptor Blocking Reagent from Miltenyi at 1:200 in antibody diluent (3% BSA in $1 \times$ PBS), adding 100 μL of the solution to each slide for 1 h at room temperature. After Fc-blocking, slides could be stained overnight with any of the individual stain rounds described in Tables EV2 and EV3, diluting

**Table 2. Patient clinical information.**

| Diagnosis | Tissue ID | Age | Sex | Disease duration (years) | Treatment |
|---|---|---|---|---|---|
| RA | JRP091_3 | 70 | F | 35 | Hydroxychloroquine/Plaquenil and Naproxen |
| RA | JRP099_3 | 31 | F | 21 | Prednisolone and Naproxen |
| RA | JRP112_6 | 35 | F | 17 | Sulfasazine/Salazopyrine/Prednisolone/NSAID |
| RA | JRP118_2 | 66 | M | 20 | Methotrexate/NSAID |
| RA | JRP122_6 | 60 | M | / | Methotrexate, Salazopyrine, Golimumab/Simponi and Prednisolone when needed |
| RA | JRP129_2 | 72 | F | 44 | Rituximab/Mabthera |
| RA | JRP136_6 | 56 | F | 36 | Prednisolone |
| RA | JRP139d_6 | 67 | M | 7 | Prednisolone and Tocilizumab (stopped 2 weeks prior to biopsy) |
| RA | JRP141_3 | 58 | F | / | / |
| OA | JRP144_3 | 74 | F | 9 | / |
| OA | S00293087_3 | 70 | F | / | / |
| OA | 97SM_15 | 64 | M | / | / |
| OA | 47SM_14 | 67 | M | / | / |
| OA | 83SM_14 | 72 | M | / | / |
| OA | S00292623_3 | 69 | F | / | / |
| OA | S00292477_3 | 88 | F | / | / |

Patient metadata available for each of the biopsy sections stained (Tissue ID) via multiplexed immunofluorescence.

the antibodies in 100 μL of antibody diluent per slide. Staining rounds were acquired on the CellDIVE system, before decoverslipping once more, and proceeding with fluorophore bleaching. For fluorophore bleaching, slides were plunged into dye inactivation solution twice for 15 min, before being washed three times in 1 × PBS for 5 min each. Slides could then be coverslipped again, and imaged. To stain for the full panels described in Tables EV2 and EV3, slides were cycled through the staining and bleaching rounds described above. Analyses of CellDIVE images were performed in QuPath (Bankhead et al, 2017). Antibodies were validated against their respective isotype controls to ensure specificity of binding (Table EV4).

### Confocal immunofluorescence—mouse tissue

Mouse knees from the Antigen-Induced Arthritis (AIA) model, run and processed as previously described (Zec et al, 2023), were stained in this study. All experimental C57BL/6 J mice were between 8 and 12 weeks old. The animals were bred and maintained in specific pathogen–free conditions in the Kennedy Institute of Rheumatology animal facility. All experiments were approved by the Clinical Medicine Animal Welfare and Ethical Review Board and the UK Home Office, in accordance with the 1986 Animals (Scientific Procedures) Act. Mouse knees were excised and fixed in PLP buffer at 4 °C overnight. After this, the knees were washed with 1 × PBS and decalcified in 10 mL of 0.5 M EDTA (pH <8) for 6 days at 4 °C, with the solution being exchanged twice. Knees were then placed in 10 mL of 30% sucrose for 24 h at 4 °C, embedded in OCT, and kept in a −80 °C freezer for long-term storage. In total, 10 μm-thick sections were cut on the Leica CM3050 cryostat at −20 °C, and slide-bound sections were placed in a −20 °C freezer. For immunofluorescent staining, slides were removed from the freezer and placed vertically in a staining dish for 1 h to air dry, followed by being placed in an oven at 65 °C for 20 min. Once dried and cooled, the

sections were circled with a PAP pen to create a hydrophobic barrier around the tissue sections. Subsequently, the slides were rehydrated with 1 × PBS for 5 min before permeabilisation with 400 μl of PBT (PBS, 0.3% Triton-X) for 5 min per slide. Blocking of nonspecific binding sites was achieved by adding 400 μL of muIF blocking solution to each slide and incubating at room temperature for 1 h in a humidified chamber. The slides were then stained by adding 400 μL of antibodies, listed in Table EV5, diluted in muIF staining solution to each slide and placed them overnight at 4 °C in a humidified chamber. The slides were then washed again three times in 1 × PBS for 5 min each, before adding 400 μL of 2 μM SyTox Blue dye in PBT per slide and incubating them for 20 min at room temperature. Slides were finally rinsed with dH₂O, dried with Kimwipes, and coverslipped by adding a drop of FluorSave mounting medium per section on the slide. Slides were lastly left to dry in the dark at room temperature overnight. Slides were then imaged using the ZEISS 980 Confocal microscope.

## Cell culture

THP-1 cells (ATCC TIB-202) or Jurkat cells (ATCC TIB-152) were cultured in suspension with RPMI 1640 medium (Gibco, 11875093) containing 10% foetal bovine serum (FBS) (Gibco, A5256801) and 1% Pen/Strep (P/S) (Gibco, 15140122). Cells were mycoplasma tested every 6 months to ensure representative cell biology and phenotypes.

## Migration assays

Transwell migration assays were performed using 24-well inserts with a pore size of 5 μm (Corning, 3421). The upper surface of the insert was coated at 37 °C for 2 h with 50 μl of either 10 μg/ml fibronectin (Abcam, ab80021) or collagen VI (Abcam, ab7538) alone, or with a combination of both to give a ratio of 1:1, 1:2, 1:4

or 1:16, where the fibronectin concentration was kept constant. The bottom reservoir was filled with 600 µl medium (RPMI 1640, 10% FBS, 1% P/S) and $1 \times 10^5$ THP-1 cells or $2.5 \times 10^5$ Jurkat cells (100 µl/well) in RPMI 1640, 0.5% FBS, 1% P/S medium were added to the top of the inserts. As a control to assess random non-directed cell migration the bottom reservoir was filled with 600 µl medium containing 0.5% FBS. After 3 h of incubation at 37 °C cell confluency in the bottom reservoir was quantified using the Incucyte Live-Cell Analysis System (Sartorius).

## Data availability

This study includes no data deposited in external repositories.

The source data of this paper are collected in the following database record: biostudies:S-SCDT-10_1038-S44320-025-00149-7.

## Peer review information

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

## Acknowledgements

This work was supported by funding from the Kennedy Trust for Rheumatology Research, the MRC CRUK, and the Research into Inflammatory Arthritis Centre Versus Arthritis UK (grant no. 22072).

## Author contributions

**Jean-Baptiste Richard**: Conceptualisation; Data curation; Formal analysis; Supervision; Validation; Investigation; Visualisation; Methodology; Writing—original draft; Writing—review and editing. **Anna Hoyle**: Data curation; Formal analysis; Investigation; Methodology; Writing—review and editing. **Molly Bower**: Formal analysis; Writing—review and editing. **Shihong Wu**: Formal analysis; Writing—review and editing. **Leia Worthington**: Formal analysis; Writing—review and editing. **Sarah Davidson**: Conceptualisation; Resources;

Data curation; Software; Formal analysis; Investigation; Methodology; Writing —review and editing. **Zofia Varyova**: Formal analysis; Writing—review and editing. **Caroline Morrell**: Formal analysis; Writing—review and editing. **Mathilde Pohin**: Resources; Methodology; Writing—review and editing. **Barbora Schonfeldova**: Resources; Methodology; Writing—review and editing. **Zhi Yi Wong**: Formal analysis; Writing—review and editing. **Lucy MacDonald**: Resources; Methodology; Writing—review and editing. **Mariola Kurowska-Stolarska**: Resources; Methodology; Writing—review and editing. **Stephanie G Dakin**: Resources; Methodology; Writing—review and editing. **Irina Udalova**: Resources; Supervision; Writing—review and editing. **Calliope A Dendrou**: Conceptualisation; Resources; Data curation; Software; Formal analysis; Methodology; Writing—review and editing. **Anja Schwenzer**: Conceptualisation; Data curation; Formal analysis; Supervision; Methodology; Writing—review and editing. **Christopher D Buckley**: Conceptualisation; Supervision; Project administration; Writing—review and editing. **Kim S Midwood**: Conceptualisation; Resources; Data curation; Supervision; Funding acquisition; Writing—original draft; Project administration; Writing—review and editing.

Source data underlying figure panels in this paper may have individual authorship assigned. Where available, figure panel/source data authorship is listed in the following database record: biostudies:S-SCDT-10_1038-S44320-025-00149-7.

## Disclosure and competing interests statement

CDB conflicts are: Consultancy (Paid): GSK, AbbVie, Takeda, CICOR Roche, Janssen Innovative Medicines. Investigator Initiated Grants (funded to University of Oxford) GSK, Janssen Innovative medicines. Founding shares: Mestag Therapeutics. The remaining authors declare no competing interests.

