## [Peer Review File · Molecular Systems Biology]

Synovial matrix turnover controls immune cell spatial patterning in inflammation resolution.

Jean-Baptiste Richard, Anna Hoyle, Molly Bower, Shihong Wu, Leia Worthington, Sarah Davidson, Zofia Varyova, Caroline Morrell, Mathilde Pohin, Barbora Schonfeldova, Zhi Wong, Lucy MacDonald, Mariola Kurowska-Stolarska, Stephanie Dakin, Irina Udalova, Calliope Dendrou, Anja Schwenzer, Christopher Buckley, and Kim Midwood

Corresponding author(s): Kim Midwood (kim.midwood@kennedy.ox.ac.uk)

Review Timeline:

Submission Date:	4th Mar 25
Editorial Decision:	16th Apr 25
Revision Received:	10th Jul 25
Editorial Decision:	14th Aug 25
Revision Received:	28th Aug 25
Accepted:	2nd Sep 25

Editor: Jingyi Hou

Transaction Report:

16th Apr 2025

Manuscript Number: MSB-2025-12951-T

Title: Synovial matrix turnover controls immune cell spatial patterning in inflammation resolution.

Author: Jean-Baptiste Richard

Kim Midwood

Anna Hoyle

Molly Bower

Shihong Wu

Leia Worthington

Zofia Varyova

Caroline Morrell

Mathilde Pohin

Barbora Schonfeldova

Zhi Wong

Lucy MacDonald

Mariola Kurowska-Stolarska

Stephanie Dakin

Irina Udalova

Anja Schwenzer

Christopher Buckley

Dear Dr. Midwood,

Thank you for submitting your work to Molecular Systems Biology. We have now heard back from the three reviewers who agreed to evaluate your manuscript. As you will see from the reports below, the reviewers find the study potentially interesting. They raise, however, a series of concerns, which should be convincingly addressed in a major revision.

The reviewers' recommendations are relatively clear, so there is no need to reiterate the points listed below. All the issues raised by the reviewers need to be satisfactorily addressed. In particular, Reviewer #2's concerns regarding the presentation and potential overstatements should be carefully addressed. While we do not recommend dividing the data into two separate manuscripts, we strongly encourage you to improve the organization of the paper to enhance the clarity and presentation of the data.

As you may already know, our editorial policy allows in principle a single round of major revision, and it is therefore essential to provide responses to the reviewers' comments that are as complete as possible. Please feel free to contact me in case you would like to discuss in further detail any of the issues raised by the reviewers.

On a more editorial level, we would ask you to address the following issues:

- Please provide a .docx formatted version of the manuscript text (including legends for main figures, EV figures and tables). Please make sure that the changes are highlighted to be clearly visible.
- Please provide individual production quality figure files as .eps, .tif, .jpg (one file per figure).
- Please provide a .docx formatted letter INCLUDING the reviewers' reports and your detailed point-by-point responses to their comments. As part of the EMBO Press transparent editorial process, the point-by-point response is part of the Review Process File (RPF), which will be published alongside your paper.
- Please note that all corresponding authors are required to supply an ORCID ID for their name upon submission of a revised manuscript.
- We replaced Supplementary Information with Expanded View (EV) Figures and Tables that are collapsible/expandable online (see examples in <http://msb.embopress.org/content/11/6/812>). A maximum of 5 EV Figures can be typeset. EV Figures should be cited as 'Figure EV1, Figure EV2' etc... in the text and their respective legends should be included in the main text after the legends of regular figures.

Additional Tables/Datasets should be labeled and referred to as Table EV1, Dataset EV1, etc. Legends have to be provided in a separate tab in case of .xls files. Alternatively, the legend can be supplied as a separate text file (README) and zipped together with the Table/Dataset file.

For the figures and tables that you do NOT wish to display as Expanded View figures, they should be bundled together with their

legends in a single PDF file called *Appendix*, which should start with a short Table of Content. Each legend should be below the corresponding Figure/Table in the Appendix. Appendix figures and tables should be referred to in the main text as: "Appendix Figure S1, Appendix Figure S2, Appendix Table S1" etc. See detailed instructions regarding expanded view here: <https://www.embopress.org/page/journal/17444292/authorguide#expandedview>.

-Before submitting your revision, primary datasets (and computer code, where appropriate) produced in this study need to be deposited in an appropriate public database (see [http://msb.embopress.org/authorguide - dataavailability](http://msb.embopress.org/authorguide-dataavailability) <https://www.embopress.org/page/journal/17444292/authorguide#dataavailability>). Please remember to provide a reviewer password if the datasets are not yet public. The accession numbers and database should be listed in a formal "Data Availability" section (placed after Materials & Method) that follows the model below (see also <https://www.embopress.org/page/journal/17444292/authorguide#dataavailability>). Please note that the Data Availability Section is restricted to new primary data that are part of this study.

Data availability

-At EMBO Press we ask authors to provide source data for the main manuscript figures. You will receive a separate email with instructions for providing source data with your revised manuscript, including how to upload and organize the files.

- Our journal encourages inclusion of *data citations in the reference list* to directly cite datasets that were re-used and obtained from public databases. Data citations in the article text are distinct from normal bibliographical citations and should directly link to the database records from which the data can be accessed. In the main text, data citations are formatted as follows: "Data ref: Smith et al, 2001". In the Reference list, data citations must be labeled with "[DATASET]". A data reference must provide the database name, accession number/identifiers and a resolvable link to the landing page from which the data can be accessed at the end of the reference. Further instructions are available at .

- We updated our journal's competing interests policy in January 2022 and request authors to consider both actual and perceived competing interests. Please review the policy <https://www.embopress.org/competing-interests> and update your competing interests if necessary.

Please use the heading "Disclosure statement and competing interests".

- All Materials and Methods need to be described in the main text using our 'Structured Methods' format. According to this format, the Methods section includes a Reagents and Tools Table (listing key reagents, experimental models, software and relevant equipment and including their sources and relevant identifiers) followed by a Methods and Protocols section describing the methods, ideally using a step-by-step protocol format. The aim is to facilitate adoption of the methodologies across labs. Please download and fill our Reagents and Tools Table template (.docx), which you can find in our author guidelines: <https://www.embopress.org/page/journal/17444292/authorguide#structuredmethods>.

An example of a Method paper with Structured Methods can be found here: <https://www.embopress.org/doi/10.15252/msb.20178071>.

-Regarding data quantification:

Please ensure to specify the name of the statistical test used to generate error bars and P values, the number (n) of independent experiments (please specify technical or biological replicates) underlying each data point and the test used to calculate p-values in each figure legend. Discussion of statistical methodology can be reported in the materials and methods section, but figure legends should contain a basic description of n, P and the test applied.

Graphs must include a description of the bars and the error bars (s.d., s.e.m.).

- Please provide a "standfirst text" summarizing the study in one or two sentences (approximately 250 characters, including space), three to four "bullet points" highlighting the main findings and a "synopsis image" (550px width and 400-600 px height, PNG format) to highlight the paper on our homepage.

Here are a couple of examples:

<https://www.embopress.org/doi/10.15252/msb.20199356>

<https://www.embopress.org/doi/10.15252/msb.20209475>

<https://www.embopress.org/doi/10.15252/msb.209495>

When you resubmit your manuscript, please download our CHECKLIST (<https://www.embopress.org/pb-assets/embosite/EMBO%20Press%20Author%20Checklist-1642513524327.xlsx>) and include the completed form in your submission.

Please note that the Author Checklist will be published alongside the paper as part of the transparent process (<https://www.embopress.org/page/journal/17444292/authorguide#transparentprocess>).

If you feel you can satisfactorily deal with these points and those listed by the referees, you may wish to submit a revised version of your manuscript. Please attach a covering letter giving details of the way in which you have handled each of the points raised by the referees. A revised manuscript will be once again subject to review and you probably understand that we can give you no guarantee at this stage that the eventual outcome will be favorable.

I look forward to receiving a revised manuscript soon.

Yours sincerely,
Jingyi

Jingyi Hou, PhD
Senior Editor
Molecular Systems Biology

We realize that it is difficult to revise to a specific deadline. In the interest of protecting the conceptual advance provided by the work, we recommend a revision within 3 months (15th Jul 2025). Please discuss the revision progress ahead of this time with the editor if you require more time to complete the revisions. Use the link below to submit your revision:

IMPORTANT: When you send your revision, we will require the following items:

1. the manuscript text in LaTeX, RTF or MS Word format
 2. a letter with a detailed description of the changes made in response to the referees. Please specify clearly the exact places in the text (pages and paragraphs) where each change has been made in response to each specific comment given
 3. three to four 'bullet points' highlighting the main findings of your study
 4. a short 'blurb' text summarizing in two sentences the study (max. 250 characters)
 5. a 'thumbnail image' (550px width and max 400px height, Illustrator, PowerPoint or jpeg format), which can be used as 'visual title' for the synopsis section of your paper.
 6. Please include an author contributions statement after the Acknowledgements section (see <https://www.embopress.org/page/journal/17444292/authorguide>)
 7. Please complete the CHECKLIST available at (<https://bit.ly/EMBOPressAuthorChecklist>).
- Please note that the Author Checklist will be published alongside the paper as part of the transparent process (<https://www.embopress.org/page/journal/17444292/authorguide#transparentprocess>).

See also figure legend guidelines: <https://www.embopress.org/page/journal/17444292/authorguide#figureformat>

9. Please note that corresponding authors are required to supply an ORCID ID for their name upon submission of a revised manuscript (EMBO Press signed a joint statement to encourage ORCID adoption). (<https://www.embopress.org/page/journal/17444292/authorguide#editorialprocess>)

Currently, our records indicate that the ORCID for your account is 0000-0002-8813-2977.

Link Not Available

11. Include a Reagents and Tools Table as part of the Methods section, which can be downloaded from our author guidelines (<https://www.embopress.org/page/journal/17444292/authorguide#structuredmethods>)

*** PLEASE NOTE *** As part of the EMBO Press transparent editorial process initiative (see our Editorial at <https://dx.doi.org/10.1038/msb.2010.72>), Molecular Systems Biology publishes online a Review Process File with each accepted

manuscripts. This file will be published in conjunction with your paper and will include the anonymous referee reports, your point-by-point response and all pertinent correspondence relating to the manuscript. If you do NOT want this File to be published, please inform the editorial office at contact@molsystbiol.org within 14 days upon receipt of the present letter.

Reviewer #1:

The article explores the role of the extracellular matrix (ECM) in regulating the spatial distribution of immune cells in synovial inflammation, particularly the function of Collagen VI in the resolution of inflammation. This is a relatively novel research direction that provides new perspectives for understanding the pathological mechanisms of immune-mediated inflammatory diseases such as rheumatoid arthritis (RA).

This study combines single-cell RNA sequencing (scRNAseq) and multiplex immunofluorescence to deeply analyze the ECM composition and cellular subpopulations in RA synovial tissue. It reveals the roles of different fibroblast subpopulations in defining the synovial microenvironment and the specific mechanisms of Collagen VI in inflammation resolution, including its inhibitory effect on the migration of immune cells and the paradoxical phenomenon where monocytes degrade it to facilitate the entry of immune cells. Overall, this research provides a comprehensive and in-depth analysis. I have few comments the authors might want to consider/address:

1. Appropriately expand the sample size of the study to enhance the generality and representativeness of the research findings.
2. Further explore the specific mechanisms of Collagen VI in synovial inflammation, including its interactions with other cell types or ECM components, as well as its direct impact on immune cell functions. Cell-cell interaction analysis can be added using single cell RNA sequencing data.
3. The TGF β signaling pathway can inhibit inflammation in certain scenarios. For instance, it promotes the differentiation of naïve CD4+ T cells into Tregs or suppresses the production of pro-inflammatory factors. In Result 3, synovial samples in the active inflammation state exhibit a higher expression of TGF β . However, the TGF β signaling pathway does not seem to exhibit its anti-inflammatory effect here. You provided an explanation for this issue in the Discussion section, but it is not entirely convincing to me. Could you provide more experiments or literature support?

Reviewer #2:

Richard et al are presenting a manuscript on matrix turnover in the inflamed tissue of patients with autoimmune disease and patients with degenerative joint disease. They have used the model system of rheumatoid arthritis. The title touches on several poorly understood areas in chronic inflammation: Matrix turnover, immune cell spatial patterning, inflammation resolution. "Synovial matrix turnover controls immune cell spatial patterning in inflammation resolution"

The authors find:

Figure 1. Matrisomal gene set enrichment of synovial cell subsets in rheumatoid arthritis

Figure 2. Canonical fibroblast subsets are defined by distinct matrix expression profiles

Figure 3. Unique and common matrisomal differential expression signatures between rheumatoid arthritis and frozen shoulder

Figure 4. Expression dysregulation of collagen VI and its processing machinery is a key feature of Rheumatoid Arthritis disease activity

Fig. 5: Perivascular collagen VI deposition associates with inflammation resolution in human RA, human OA, and mouse antigen-induced arthritis

Fig. 6: Collagen VI deposition patterns create "dark" zones

Fig. 7: Tissue monocyte infiltration is degradation associated and the fragments from this degradation are detectable in tissue.

The authors conclude:

"Our work reveals a novel mechanism of cell immigration regulation in RA. It provides new evidence to support the use of collagen VI fragments as biomarkers of tissue-level disease activity, and for exploring stromal targets as novel therapeutic avenues."

General remarks:

1. Most of this article consists of comparative analysis of publicly available data sets and might fit much better into a review article.
2. Reorganization of the paper could help the reader to understand much better which data are new and which data can support the conclusions. It may also be an option to separate the comparative data analysis of publicly available data sets and the

experimental data into two manuscripts.

3. At times it is difficult to connect the data presented to the conclusions. Which novel mechanism of cell-migration regulation was discovered? What is the evidence supporting collagen VI fragments as biomarkers of disease activity? Which novel therapeutic avenues should be explored. It would also be important to place the data into the context of previous clinical trials in which fibroblasts were targeted, but with not much success.

4. The authors emphasize the enormous gaps in the treatment options for patients with immune-mediated disease (first paragraph: "IMIDs are often plagued by poor response rates to available treatments, and a complete lack of curative therapeutic avenues"), yet throughout the manuscript that talk about "remission".

What is the evidence for "remission"? The authors may want to consider using the term "low disease activity" instead.

In the first paragraph of the Results section, they introduce "a powerful dataset" that they extensively use throughout the manuscript: 4 patients with active disease and 3 patients with disease in remission. How robust is this dataset to serve as a foundation for major conclusions or the development of new treatments?

Major points:

5. Fig. 1 and 2: By using publicly available data sets, the authors find that fibroblasts produce matrix proteins and macrophages produce matrix-digestive enzymes. That is not a surprise.

Could these data be moved to the Supplementary data?

6. Fig. 3: By comparing rheumatoid arthritis and frozen shoulder, the authors find that there is a "cross-disease" remission signature.

This will make is very difficult to develop matrix specific biomarkers. Shoulder disease and other degenerative joint diseases are frequent in the population and will then produce biomarkers that are indistinguishable from the signature of the autoimmune joint disease?

This should subtract considerably from the value of such matrix biomarkers.

7. Fig.3/Fig.4:

The heatmap in panel C suggests that while there may be transcriptional differences for collagens comparing autoimmune versus degenerative disease, differences in protein abundance are minimal. This questions the biologic significance of the transcriptomic studies.

8. Fig. 3/Fig.4:

Panels D and E: three data points are presented, in some panels with considerable variation.

There are not enough data points to provide confidence to draw conclusions.

9. Fig. 5:

Staining of tissue sections demonstrates that the two disease conditions, the immune-mediated condition and the degenerative condition, produce very similar distribution patterns of collagen VI. Correlations in panels D and E for the two diseases are essentially identical.

This is interesting in itself, suggesting that fibroblasts have only a limited number of response patterns.

Here, the authors may want to shift their interest to similarities instead of differences.

10. Fig. 6:

The authors are offering the concept of "dark zones", which are areas in the tissue where immune cells accumulate, and no collagen VI is deposited.

It is not easy to follow this concept.

Immune cells do not accumulate in collagen-rich tissue sites, best exemplified in lymph nodes.

It is to be expected that immune cells are clustered in collagen free tissue niches.

Again, this figure emphasizes the similarities between inflammation induced by an immune-mediated process and a degenerative process.

11. Fig. 7:

The tissue stain shows a low signal for collagen VI. May be, a better example could be chosen.

Minor points:

Please note that there are two Figures 3.

Reviewer #3:

Summary:

Here, Richard et al. reanalyze published single cell RNA-seq datasets and generate multiplexed immunofluorescence imaging

data to describe the role of collagen VI in rheumatoid arthritis remission. Collagen VI increased in remission compared to active RA, concomitant with decreased degradation by MMP enzymes. Within synovial tissue slices, collagen VI and multiple immune cell types showed mutual spatial exclusion, with the exception of monocytes, which the authors suggest degrade collagen VI.

General remarks:

The role of fibroblasts and the ECM over the course of autoimmune diseases has been undercharacterized, and relapses in autoimmune diseases remain a major clinical problem. This work applies relatively new methods (data from single-cell sequencing, multiplexed IF) to provide interesting insights with potential clinical relevance across multiple diseases. However, a few points require clarification before publication.

Major points:

Parts of this paper overlap with a separate bioRxiv preprint by some of the same authors (Nisa et al., 2024). For example the transwell assay described in the preprint is very similar to that performed in here: "Compared to wells without a collagen coating, there was a significant decrease in the migration of monocytes ($p < 0.01$), CD4+ and CD8+ T cells ($p < 0.05$) through collagen VI-coated wells, as well as significantly decreased CD4+ and CD8+ T cell migration through collagen I-coated wells ($p < 0.05$) (Figure 4f). These data suggest the presence of collagens I and VI restrict migration of specific leukocytes through ECs and into the tissue." Overlaps (and inconsistencies regarding the role of collagen I) with this paper should be explicitly discussed.

Subhead 4: Does RA in remission resemble OA? Can the authors quantify with their transcriptomics data that these states are more similar to each other than to active RA? This is important if using the proteomics comparison between RA and OA to infer differences in remission.

Subhead 4: Where is the data and quantification to show that collagen VI transcripts are upregulated in RA?

"Despite overexpression in active RA and RA compared to remission and OA respectively, collagen VI was significantly less abundant in RA compared to OA": expression and abundance can refer to either transcript or protein levels. This sentence should be rewritten to clarify.

Subhead 4: it's not clear why the authors follow up on collagen VI rather than others that differ between RA and OA. Specifically, the authors note that COL2A1 and COL15A1 are higher in OA than RA, similar to COL6A1. Why were these (and other collagen VI genes) not included in the IF panel, and is there any data to suggest that they are less important?

"collagen VI degrading proteases MMP2 and MMP9": a potentially misleading description, as MMP2 and MMP9 are known to degrade other collagen types as well.

Subhead 5: It's unclear why COL6A1 quantification is reported in the text for the perivascular region and not across the whole tissue, as shown in Figure S7, except that the MerTK+ comparison becomes significant (are p values corrected for multiple testing?). The authors also focus in the text on the correlation between COL6A1+ area and MerTK+ macrophages, but Figure S7 shows positive correlations with unexpected cell types, including T cells. How do they explain this?

Subhead 6-7: Can the authors try a transwell assay with CD14+ monocytes to support their claim that these cells are able to migrate on collagen VI, in contrast to T cells and monocytic THP1 cells? Do THP1 cells not express the same MMPs as other monocytes?

Minor points:

Subhead 2: Does IGFBP7 stain endothelial cells, mural cells, or others around the vasculature?

Subhead 4: Can the authors suggest any potential mechanisms for the upregulation of collagen VI transcripts in active RA?

Subhead 7: Do the authors have any citations or validation data for the CHP peptide staining?

For clarification and readability:

Introduction: Explain the function of MMPs at first mention

Subhead 1: What are the main functional differences among matrisome gene categories?

Subhead 2:

"These subsets were originally defined through a multicentre international effort to map the synovium, as a pioneer study bridging the strengths of academia and industry" - is this description necessary? It distracts from the scientific results in the present paper

"SC-F1", etc. are jargon for anyone not well-versed in the previous study

Subhead 3: Gene category nomenclature is inconsistent across paragraphs (e.g. "remission-downregulated" vs. "active inflammation")

Figure 6 caption: note cell types associated with each marker

Discussion: is the Nordic Bioscience data referred to from patients pre-treatment or early in treatment?

"Figure 4" is mislabelled "Figure 3" in the caption

REVIEWER COMMENTS

We thank the reviewers for their constructive comments on our manuscript and we appreciate the opportunity to revise the paper in response. We have clarified the presentation and interpretation of the existing data, and we have performed new analyses and added new data to address the comments where possible. We think that this has strengthened the message and impact of the paper, and we look forward to your comments on the revised version.

Editor

Reviewer #2's concerns regarding the presentation and potential overstatements should be carefully addressed. While we do not recommend dividing the data into two separate manuscripts, we strongly encourage you to improve the organization of the paper to enhance the clarity and presentation of the data.

Please see our response to reviewer 2 below in which we address these points.

Reviewer #1:

The article explores the role of the extracellular matrix (ECM) in regulating the spatial distribution of immune cells in synovial inflammation, particularly the function of Collagen VI in the resolution of inflammation. This is a relatively novel research direction that provides new perspectives for understanding the pathological mechanisms of immune-mediated inflammatory diseases such as rheumatoid arthritis (RA). This study combines single-cell RNA sequencing (scRNAseq) and multiplex immunofluorescence to deeply analyze the ECM composition and cellular subpopulations in RA synovial tissue. It reveals the roles of different fibroblast subpopulations in defining the synovial microenvironment and the specific mechanisms of Collagen VI in inflammation resolution, including its inhibitory effect on the migration of immune cells and the paradoxical phenomenon where monocytes degrade it to facilitate the entry of immune cells. Overall, this research provides a comprehensive and in-depth analysis. I have few comments the authors might want to consider/address:

1. Appropriately expand the sample size of the study to enhance the generality and representativeness of the research findings.

We have performed a number of additional analyses to expand the sample size of the study and to validate key research findings:

We have added new data from two recently published transcriptomic studies, one comprising 88 RA patient synovial biopsies, and one that allows us for the first time to include healthy control synovial cells. These analyses are shown in new Appendix Figures S6 and S10 and discussed in the text on pages 10, 11 and 14.

We have added data from new degradomic analysis of two further independent proteomic datasets to support our original mass spec analysis. These data are shown in new Figure 4F and G and new Appendix figure S7, and discussed in the text on page 11 and 12.

Finally, as also recommended by reviewer 2, we have better emphasized on pages 22-25 our finding that immune-rich, COL6-poor, dark zones are detected not only in different diseases within the same tissue type (synovium, RA and OA), but are also found across different tissues with different types of pathology (intestine with IBD, head and neck tumors), indicating that this means of regulating immune ingress is a well conserved, generalizable phenomenon.

2. Further explore the specific mechanisms of Collagen VI in synovial inflammation, including its interactions with other cell types or ECM components, as well as its direct impact on

immune cell functions. Cell-cell interaction analysis can be added using single cell RNA sequencing data.

We performed an interactome analysis for COL6A1, A2 and A3 identifying a large number of known and predicted binding partners. Pathway analysis of these interactions reveals potential roles for COL6 in matrix binding and remodelling and TGF β signalling, as well as immune cell migration, cell survival and modulation of cell proliferation. Amongst the high confidence, experimentally validated, binding partners we identified both cellular and matrix interactors. Expression of each of 47 cellular receptors was detected in synovial tissue from two independent transcriptomic data sets, with some enriched in immune cells and some in stromal cells. Amongst these synovial cell receptors for COL6, several were regulated across disease status, including T cell activation markers and myeloid cell adhesion molecules. These data are shown in new Supplementary Table 2, new Appendix figures 9, 10 and 11 and discussed in the text as potential mechanisms by which COL6 deposition in the synovium may affect immune cell function on pages 13 and 14.

3. The TGF β signaling pathway can inhibit inflammation in certain scenarios. For instance, it promotes the differentiation of naïve CD4+ T cells into Tregs or suppresses the production of pro-inflammatory factors. In Result 3, synovial samples in the active inflammation state exhibit a higher expression of TGF β . However, the TGF β signaling pathway does not seem to exhibit its anti-inflammatory effect here. You provided an explanation for this issue in the Discussion section, but it is not entirely convincing to me. Could you provide more experiments or literature support?

We have expanded our discussion of this interesting paradox and added further literature support in the text on pages 20-21.

Reviewer #2:

Richard et al are presenting a manuscript on matrix turnover in the inflamed tissue of patients with autoimmune disease and patients with degenerative joint disease. They have used the model system of rheumatoid arthritis. The title touches on several poorly understood areas in chronic inflammation: Matrix turnover, immune cell spatial patterning, inflammation resolution.

"Synovial matrix turnover controls immune cell spatial patterning in inflammation resolution"

The authors find:

Figure 1. Matrisomal gene set enrichment of synovial cell subsets in rheumatoid arthritis

Figure 2. Canonical fibroblast subsets are defined by distinct matrix expression profiles

Figure 3. Unique and common matrisomal differential expression signatures between rheumatoid arthritis and frozen shoulder

Figure 4. Expression dysregulation of collagen VI and its processing machinery is a key feature of

Rheumatoid Arthritis disease activity

Fig. 5: Perivascular collagen VI deposition associates with inflammation resolution in human RA, human OA,

and mouse antigen-induced arthritis

Fig. 6: Collagen VI deposition patterns create "dark" zones

Fig. 7: Tissue monocyte infiltration is degradation associated and the fragments from this degradation are detectable in tissue.

The authors conclude:

"Our work reveals a novel mechanism of cell immigration regulation in RA. It provides new evidence to support the use of collagen VI fragments as biomarkers of tissue-level disease

activity, and for exploring stromal targets as novel therapeutic avenues."

General remarks:

1. Most of this article consists of comparative analysis of publicly available data sets and might fit much better into a review article.

The rise of 'omics technologies has resulted in a wealth of publicly available data from unique human clinical samples and in vivo studies. However, often only a fraction of the information contained within these large datasets is examined in detail. Whilst we use existing datasets in this paper, we perform new analyses of these data, which has generated new information. For example, detailed annotation of the cell types expressing matrix genes in each of these datasets has not been reported, this is the first cross-species and cross-disease matrix comparison, and interrogating matrix protein abundance versus degradation in the proteomic datasets has not been performed previously. In this paper we are not summarizing data that has already been published, rather we are using existing resources to generate novel data, and this approach has uncovered an unexpected role for matrix degradation in immune resolution.

2. Reorganization of the paper could help the reader to understand much better which data are new and which data can support the conclusions. It may also be an option to separate the comparative data analysis of publicly available data sets and the experimental data into two manuscripts.

Based on the recommendation of the editor we have not split the paper into two, but we have clarified in the figure legends which parts of the dataset analysis are already published (for example, cell annotation and clustering) and which parts are new (for example, cell re-clustering, matrixome and degradome interrogation).

3. At times it is difficult to connect the data presented to the conclusions. Which novel mechanism of cell-migration regulation was discovered? What is the evidence supporting collagen VI fragments as biomarkers of disease activity? Which novel therapeutic avenues should be explored. It would also be important to place the data into the context of previous clinical trials in which fibroblasts were targeted, but with not much success.

We have moderated these statements in the abstract on page 2, and other statements throughout the text, and we have added to the discussion a section that places these data in the context of previous clinical trials targeting fibroblasts in RA on page 19.

4. The authors emphasize the enormous gaps in the treatment options for patients with immune-mediated disease (first paragraph: "IMIDs are often plagued by poor response rates to available treatments, and a complete lack of curative therapeutic avenues"), yet throughout the manuscript that talk about "remission". What is the evidence for "remission"? The authors may want to consider using the term "low disease activity" instead.

We have clarified in the introduction on pages 3 and 4 the currently accepted clinical definition of remission and added literature citing that this is usually a temporary state.

In the first paragraph of the Results section, they introduce "a powerful dataset" that they extensively use throughout the manuscript: 4 patients with active disease and 3 patients with disease in remission. How robust is this dataset to serve as a foundation for major conclusions or the development of new treatments?

We have moderated our description on page 5 as stated below. We have also added data from two further human transcriptomic datasets that extends our analysis of data from gene triaging (new supplementary Figure 6) and we discuss this in the text on pages 10-11.

“The field of RA has also benefited from the development of **valuable** datasets including a scRNAseq dataset compiling whole synovial cells from 4 patients with active RA and 3 patients with RA in remission...”

Major points:

5. Fig. 1 and 2: By using publicly available data sets, the authors find that fibroblasts produce matrix proteins and macrophages produce matrix-digestive enzymes. That is not a surprise.

Could these data be moved to the Supplementary data?

With these figures we wanted to emphasize that to fully understand the extracellular matrix of any tissue it is important to take into account how all cells work together to shape the microenvironmental cues that cellular residents respond to. We've expanded the text in the results on page 6 to better highlight this, but we think this message might be lost if we move these data to the supplementary figures.

6. Fig. 3: By comparing rheumatoid arthritis and frozen shoulder, the authors find that there is a "cross-disease" remission signature. This will make it very difficult to develop matrix specific biomarkers. Shoulder disease and other degenerative joint diseases are frequent in the population and will then produce biomarkers that are indistinguishable from the signature of the autoimmune joint disease? This should subtract considerably from the value of such matrix biomarkers.

We have clarified in the results text on page 9 and 10 that in our cross-disease analysis, as well as identifying a signature that is commonly expressed in the remission phase of both RA and frozen shoulder, we also identify signatures that are unique to the active and remission stages of each condition, and we now better highlight the utility of each of these signatures.

7. Fig.3/Fig.4:

The heatmap in panel C suggests that while there may be transcriptional differences for collagens comparing autoimmune versus degenerative disease, differences in protein abundance are minimal. This questions the biologic significance of the transcriptomic studies.

We agree and we have added text on page 13 to clarify the different, and mutually beneficial, value of examining both transcriptomic and proteomic data sets in parallel.

8. Fig. 3/Fig.4:

Panels D and E: three data points are presented, in some panels with considerable variation. There are not enough data points to provide confidence to draw conclusions.

We have clarified the experimental design of this study: each of the 3 dots shown represents a combined figure denoting the mean abundance of every tryptic (D) and every semi-tryptic (E) peptide that was detected within the chain of both of the collagens studied (Col1A1 and A2; left panel and Col6 A1, A2 and A3; right panel) from tissue from a total of 12 RA and 12 OA patients pooled into batches of 3 (old figure 4D, E, new Appendix figure 7A, B). We have added new plots to better show the raw data underpinning these summed data. In new Appendix figure S7C and D we have plotted out all peptides detected across the collagen chains individually, and in new Figure 4D and E we have plotted all peptides from every collagen chain combined, to highlight the spread of data. In addition, we have added

analysis from two independent proteomic datasets that validate our findings (new figure 4 F and G, Appendix Figure S7 E, F) and we discuss these data on pages 11-12 of the manuscript.

9. Fig. 5:

Staining of tissue sections demonstrates that the two disease conditions, the immune-mediated condition and the degenerative condition, produce very similar distribution patterns of collagen VI. Correlations in panels D and E for the two diseases are essentially identical. This is interesting in itself, suggesting that fibroblasts have only a limited number of response patterns. Here, the authors may want to shift their interest to similarities instead of differences.

We have added text to the results on page 15 to better emphasize the implications of similar COL6 protein distribution in the two different diseases and we also clarify in the discussion the conservation of this response as a disease agnostic fibroblast response (pages 22-23, page 24-25).

10. Fig. 6:

The authors are offering the concept of "dark zones", which are areas in the tissue where immune cells accumulate, and no collagen VI is deposited. It is not easy to follow this concept.

Immune cells do not accumulate in collagen-rich tissue sites, best exemplified in lymph nodes.

It is to be expected that immune cells are clustered in collagen free tissue niches. Again, this figure emphasizes the similarities between inflammation induced by an immune-mediated process and a degenerative process.

We think the interesting finding here is the specificity of the collagen deposition or lack of deposition in the 'dark zones'. As shown in Appendix Figure S12 the ring of COL6 (but neither COL4 nor COL1) contains the CD45+ cell infiltrate, and whilst COL1 is distributed evenly throughout the dark zone, COL6 is lacking. These data fit with our proteomic data showing that COL6 but not COL1 degradation is enhanced in RA, and published data showing that myeloid-derived proteases MMP2 and MMP9 cleave COL6 but not fibrillar collagens including COL1 which instead rely on digestion by collagenases including MMPs1, 8, 13 and 14. We have tried to clarify this in the text by calling the dark zones specifically 'COL6 dark zones'. We have also added more discussion to highlight the similarities not only between immune and degenerative diseases of the joint, but indeed also in samples of intestinal tissue from people with IBD and from head and neck tumors on pages 22-25.

11. Fig. 7:

The tissue stain shows a low signal for collagen VI. May be, a better example could be chosen.

We have added a better image to Figure 7.

Minor points:

Please note that there are two Figures 3.

We have corrected the legend of the second figure 3, from figure 3 to figure 4.

Reviewer #3:

Summary:

Here, Richard et al. reanalyze published single cell RNA-seq datasets and generate

multiplexed immunofluorescence imaging data to describe the role of collagen VI in rheumatoid arthritis remission. Collagen VI increased in remission compared to active RA, concomitant with decreased degradation by MMP enzymes. Within synovial tissue slices, collagen VI and multiple immune cell types showed mutual spatial exclusion, with the exception of monocytes, which the authors suggest degrade collagen VI.

General remarks:

The role of fibroblasts and the ECM over the course of autoimmune diseases has been undercharacterized, and relapses in autoimmune diseases remain a major clinical problem. This work applies relatively new methods (data from single-cell sequencing, multiplexed IF) to provide interesting insights with potential clinical relevance across multiple diseases. However, a few points require clarification before publication.

Major points:

1. Parts of this paper overlap with a separate bioRxiv preprint by some of the same authors (Nisa et al., 2024). For example the transwell assay described in the preprint is very similar to that performed in here: "Compared to wells without a collagen coating, there was a significant decrease in the migration of monocytes ($p < 0.01$), CD4+ and CD8+ T cells ($p < 0.05$) through collagen VI-coated wells, as well as significantly decreased CD4+ and CD8+ T cell migration through collagen I-coated wells ($p < 0.05$) (Figure 4f). These data suggest the presence of collagens I and VI restrict migration of specific leukocytes through ECs and into the tissue." Overlaps (and inconsistencies regarding the role of collagen I) with this paper should be explicitly discussed.

We discuss the overlaps and differences in data generated from these different experimental models (one assay examines endothelial transmigration and one migration on fibronectin (FN) substrates designed to mimic the content of the lining and sublining stromal tissue), on page 22 and 23, as below, in the context of other published data and we highlight the biological implications of these collagen specific roles.

"Based on the location of COL6 dark zones, which arise at the vasculature basement membrane, often extending deep into the synovial (and intestinal and tumor) tissue, we aimed to model in vitro immune cell movement within tissues using FN as a migration permissive matrix molecule that is ubiquitously expressed in the stroma of all tissues including the synovium. Both monocytic and T cell lines readily migrated on FN but were unable to migrate on collagen VI alone; cell movement was also blocked on migration competent surfaces (FN) coated with collagen VI. This is consistent with studies showing that T cell migration on collagen VI substrates is reduced compared to migration on either FN or collagen I, which is caused by disrupted fibrillar actin formation required to maintain efficient motility and traction force on collagen VI that does not occur in cells interacting with FN or collagen I (Pruitt et al., 2023). These data are also consistent with data showing that collagen VI, but not collagen I or IV, significantly blocks monocyte migration in a transendothelial migration assay across activated HUVECS. In line with our data, collagen VI prevented T cell migration across the endothelial cell barrier, however, interestingly, in this assay collagen I also inhibited T cell movement across a HUVEC substrate. These data suggest that lymphocytic migration occurs via different mechanisms depending on the underlying substrate, and which have different susceptibility to collagen modulation (Nisa et al., 2024). Together, these data show that collagen VI alone does not support effective monocyte or T cell migration, and that this collagen also restricts migration of both cell types on permissive substrates, regardless of the nature of the underlying substrate. These data also demonstrate that, whilst collagen I does not impact monocyte transendothelial migration, it does prevent T cells moving across the vascular barrier. Together these data highlight a series of sequential checkpoints that monitor cell ingress into tissues at different stages."

2. Subhead 4: Does RA in remission resemble OA? Can the authors quantify with their transcriptomics data that these states are more similar to each other than to active RA? This is important if using the proteomics comparison between RA and OA to infer differences in remission.

There are similarities in synovial cell behaviour and gene expression between OA and RA in remission, and also differences. We have added new data including from healthy synovial fibroblasts to give a better sense of relative collagen expression in non-diseased, active RA, remission RA and OA in new Appendix Fig 6. We have tried to clarify in the text that we don't use OA as a proxy for remission RA but as an interesting disease state in and of itself. We have also added data showing how patterns of expression of Col6-associated remodellers in RA in remission resembles that in OA in new Appendix Figure S8, and that we use this to model collagen deposition potential in remission. Finally, we have added a summary to the discussion on pages 24-25 to clarify both disease, and disease state-, specific findings.

3. Subhead 4: Where is the data and quantification to show that collagen VI transcripts are upregulated in RA?

"Despite overexpression in active RA and RA compared to remission and OA respectively, collagen VI was significantly less abundant in RA compared to OA": expression and abundance can refer to either transcript or protein levels. This sentence should be rewritten to clarify.

We previously summarized these data in the table in Figure 4A and showed them as part of the bigger dot plot for active RA and remission in Figure 4F. We have now added a new Appendix figure S6A-H showing these data specifically for COL6 for both remission and OA so that i) the presentation of the data are clearer, ii) all data discussed in the text are shown and iii) everything can be more easily and directly found. We have also clarified this text on pages 11-12 to make clear when we refer to mRNA and when we refer to protein instead of using expression and/or abundance.

4. Subhead 4: it's not clear why the authors follow up on collagen VI rather than others that differ between RA and OA. Specifically, the authors note that COL2A1 and COL15A1 are higher in OA than RA, similar to COL6A1. Why were these (and other collagen VI genes) not included in the IF panel, and is there any data to suggest that they are less important?

We agree that the elevated protein levels of COL2A1 and COL15A1 observed in OA compared to RA synovium are interesting. We focused on COL6 here because unlike COL2 and COL15 the significantly lower protein levels of COL6 in RA directly contrasted with significantly elevated mRNA levels of COL6, suggesting that for COL6 uniquely there was a directly opposite effect between transcription and protein deposition. Further investigation of other collagens would be of interest and potentially important but we feel lies beyond the scope of this study. Lack of availability of antibodies to other COL6 chains (including COL6A2 and COL6A3) suitable for CellDive analysis precluded their inclusion in the IF panel. However, the COL6 protein is an obligate heterotrimer in which inclusion of one chain of COL6A1 is always required and which does not assemble in the absence of COL6A1, meaning that visualization of this component of the protein is a good way to visualize total COL6 protein.

5. "collagen VI degrading proteases MMP2 and MMP9": a potentially misleading description, as MMP2 and MMP9 are known to degrade other collagen types as well.

We have clarified this sentence on page 11 and as below:

“Active RA was characterised by higher relative expression of proteases *MMP2* and *MMP9*, enzymes known to cleave collagen VI, with a comparative decrease in phases of remission...”

6. Subhead 5: It's unclear why COL6A1 quantification is reported in the text for the perivascular region and not across the whole tissue, as shown in Figure S7, except that the MerTK+ comparison becomes significant (are p values corrected for multiple testing?). The authors also focus in the text on the correlation between COL6A1+ area and MerTK+ macrophages, but Figure S7 shows positive correlations with unexpected cell types, including T cells. How do they explain this?

We have corrected the p values in new appendix fig S10 (old Figure S7) for multiple testing and updated the legend and results text accordingly. We have added more comprehensive discussion of COL6A1 quantification across the whole tissue, as well the perivascular region to the text on page 15, and we also now include discussion of associations with MerTK+ macrophages and T cells. We further discuss the association with both cell types in the text on page 15, citing literature that supports known T cell interactions with COL6, alongside our new interactome data that further explores these cell-matrix interactions in the synovium.

7. Subhead 6-7: Can the authors try a transwell assay with CD14+ monocytes to support their claim that these cells are able to migrate on collagen VI, in contrast to T cells and monocytic THP1 cells? Do THP1 cells not express the same MMPs as other monocytes?

We have added further discussion around our model in the text on page 17 to clarify that we are proposing that neither monocytes nor T cells migrate on COL6, as our data show, but that monocytes are the primary source of the MMPs that degrade COL6. As such these cells become the 'pathfinders', not by virtue of their ability to migrate on COL6 where other cells' migration is inhibited, but by virtue of their capability to remove the COL6 barrier, meaning that they can then further migrate into the tissue, and T cells can follow behind in the absence of COL6.

Minor points:

8. Subhead 2: Does IGFBP7 stain endothelial cells, mural cells, or others around the vasculature?

We have added the following text to the results on page 8 to clarify where IGFBP7 staining is located:

“...and the expression of major RA-associated growth factor IGF1 (34,35) and its regulator IGFBP7 (36), with the latter interestingly specifically located around the vasculature, in particular co-staining with endothelial cells (Fig 2E).”

9. Subhead 4: Can the authors suggest any potential mechanisms for the upregulation of collagen VI transcripts in active RA?

We have added the text and new references to the discussion on pages 21 and 22 to suggest potential mechanisms for COL6 mRNA upregulation in active RA.

10. Subhead 7: Do the authors have any citations or validation data for the CHP peptide staining?

We have added the following citations to the results text on page 17 describing the design, synthesis, validation and application of the CHP peptide:

Y. Li, C.A. Foss, D.D. Summerfield, J.J. Doyle, C.M. Torok, H.C. Dietz, M.G. Pomper, S.M. Yu

Targeting collagen strands by photo-triggered triple-helix hybridization
Proc. Natl. Acad. Sci. USA, 109 (2012), pp. 14767-14772

Y. Li, D. Ho, H. Meng, T.R. Chan, B. An, H. Yu, B. Brodsky, A.S. Jun, M.S. Yu
Direct detection of collagenous proteins by fluorescently labeled collagen mimetic peptides
Bioconjugate Chem., 24 (2013), pp. 9-16

Hwang, J., et al. (2017). In Situ Imaging of Tissue Remodeling with Collagen Hybridizing Peptides. ACS Nano 11(10): 9825-9835.

Zitnay, J., *et al.* Molecular level detection and localization of mechanical damage in collagen enabled by collagen hybridizing peptides. *Nat Commun* 8, 14913 (2017).

For clarification and readability:

11. Introduction: Explain the function of MMPs at first mention

We have added the following text to the results on page 4 at first mention of MMPs:

“Overexpression of matrix-metalloproteinases (MMPs), proteases that degrade an array of matrix molecules, in the inflamed synovium has been long known (25)...”

12. Subhead 1: What are the main functional differences among matrisome gene categories?

We have added the following text on pages 5 and 6 to define the matrisomal gene categories:

“To first catalogue the synovial matrisome, gene set enrichment analysis was applied to look for specific enrichments in the expression of each matrisome gene category defined by Naba and colleagues (Fig. S1), in each cell cluster present in this dataset (Fig. 1A), using the AUCell package (31). The matrisome can be divided into two main categories, each with 3 subcategories: 1) core matrisome: which is important for the structural properties of the matrix and encompass glycoproteins, essential for cell-matrix interactions, collagens that provide the main tissue tensile strength and proteoglycans, characterized by the presence of highly charged glycosaminoglycans (GAGs) attached to a protein core, and 2) matrisome-associated proteins that play regulatory roles within the matrix and encompass ECM-affiliated proteins that share structural similarities with or are known to associate with core matrix proteins, ECM regulators, enzymes involved in ECM remodeling and cross-linking and secreted factors, molecules that bind to core ECM proteins and mediate signaling and communication. Strong enrichment across all three....”

13. Subhead 2:

"These subsets were originally defined through a multicentre international effort to map the synovium, as a pioneer study bridging the strengths of academia and industry" - is this description necessary? It distracts from the scientific results in the present paper

We have deleted this sentence.

14. "SC-F1", etc. are jargony for anyone not well-versed in the previous study

We have provided a more general introduction to these four fibroblast subtypes in the results text on page 7 to make this more accessible to all readers, and then we have kept these abbreviations for ease of labelling in the figures and in the text:

“....four fibroblast clusters defined in the first Accelerating Medicines Partnership (AMP) consortium study (Fig. S3), referred to throughout the rest of the study as AMP1 (22), and enable cross-dataset comparisons. **These four cell clusters comprise three populations of fibroblasts that locate to the sublining layer of the synovium and participate in tissue remodelling (SC-F1, SC-F3) and inflammatory processes (SC-F2), along with one population of fibroblasts that locate to the lining layer of the synovium and participate in barrier function and synovial fluid production (SC-F4)....**”

15. Subhead 3: Gene category nomenclature is inconsistent across paragraphs (e.g. "remission-downregulated" vs. "active inflammation")

We have re-phrased this section in the second paragraph of this subhead on page 9 to keep the gene nomenclature consistent:

“Key features of active inflammation that were downregulated in remission, are ECM remodelling and wound-healing responses mediated....”

16. Figure 6 caption: note cell types associated with each marker

We have added the cell types associated with each marker to the legend of figure 6:

“A, B. Visualisation of COL6A1 and its dark zones, lymphocyte markers CD4, CD8 (**T cells**), CD20 (**B cells**), and MZB1 (**plasma cells**) and myeloid and granulocyte markers CD11c (**synovial dendritic cells**), CD14 (**monocytes**), CD68 (**macrophages**), and CD66b (**neutrophils**) in RA (A) and OA (B).”

17. Discussion: is the Nordic Bioscience data referred to from patients pre-treatment or early in treatment?

We have clarified in the discussion on page 24 what the Nordic Bioscience data refer to:

“More recently, the company also described **changes in serum levels of the collagen VI degradation fragment C6M in the first 16 weeks after treatment with Tocilizumab** as a potential **indicator** of treatment efficacy (74)”

18. "Figure 4" is mislabelled "Figure 3" in the caption

We have corrected the labelling of this figure legend to Figure 4.

14th Aug 2025

Manuscript Number: MSB-2025-12951R

Title: Synovial matrix turnover controls immune cell spatial patterning in inflammation resolution.

Author: Jean-Baptiste Richard

Kim Midwood

Anna Hoyle

Molly Bower

Shihong Wu

Leia Worthington

Zofia Varyova

Caroline Morrell

Mathilde Pohin

Barbora Schonfeldova

Zhi Wong

Lucy MacDonald

Mariola Kurowska-Stolarska

Stephanie Dakin

Irina Udalova

Anja Schwenzer

Christopher Buckley

Callie Dendrou

Sarah Davidson

Dear Dr. Midwood,

Thank you for submitting the revised version of your manuscript. We have now received feedback from two reviewers who agreed to assess the revision.

As you will see below, Reviewer #1 is satisfied with the changes made. However, Reviewer #3 has raised a new concern regarding the originality of the data. Specifically, they note that a substantial portion of the analysis relies on previously published datasets and employs analytical approaches that are not sufficiently differentiated from those used in the original publication.

While this issue was not brought up during the initial round of review, we agree that the current version of the manuscript would benefit from clearer differentiation between the data analysis approach in this study and that of the original publication. In addition, the novel biological findings of your study should be more explicitly emphasized and clearly contextualized within the existing literature. Please address these issues in writing in a revised version of the manuscript.

On a more editorial level:

1. Please provide up to five keywords in the manuscript file.
2. Remove the "Author contributions" section from the manuscript file.
3. "Competing interest" should be renamed to "Disclosure and Competing Interests Statement".
4. "Acknowledgements and Funding" should be renamed to "Acknowledgements".
5. Please remove "one sentence summary" and "Subhead x" from the manuscript file.
6. Please remove the "standfirst and synopsis" from the manuscript file and upload it as a separate file. I have slightly modified the text (see attached). Please let me know if it is fine as is or if you'd like to introduce further modifications.
7. There are a few discrepancies in author names: for example, Caroline Morell appears in the manuscript text, while Caroline Morrell was entered in the online submission system; similarly, Calliope A. Dendrou is listed in the manuscript, while Callie Dendrou appears in the system. Please double-check all author names and ensure consistency across the manuscript and submission system.
8. References:
 - Please list up to 10 authors before using et al. in the reference list.

- DOIs should be included only for preprints and datasets. Please remove any DOIs that do not fall into these categories.
- For data citation, please include the original article that reported the dataset as a standard reference. You can refer to our guidelines here: <https://www.embopress.org/page/journal/17444292/authorguide#referencesformat>

9. The email addresses provided for Molly Bower (molly.bower@queens.ox.ac.uk) and Callie Dendrou (callie.dendrou@kennedy.ox.ac.uk) appear to be invalid. Please double-check and provide updated, valid email addresses for all listed authors.

10. Source Data: The source data files for Figure 5 are currently split into multiple folders. Please combine them into a single ZIP file and upload it accordingly.

11. Issues with tables

- There is a "Table 0" in the manuscript, which should be renamed to "Table 1" and called out in the main text.
- "Supplementary Table 2" should be renamed to "Table EV1" and uploaded using the correct naming convention. Please update the manuscript text accordingly.
- We recommend renaming Tables 3-6 as Tables EV2-EV5.
- Please refer carefully to our guidelines for formatting Expanded View tables:
- <https://www.embopress.org/page/journal/17444292/authorguide#expandedview>. Each EV table should be uploaded as a separate Excel (.xls) file. Each file must include a separate sheet labeled 'Legend' that contains the corresponding table legend.

12. Please add the missing callouts for Figure 4J.

13. Appendix

- On the title page of the appendix, please include the heading: "Appendix for [manuscript title]".
- Update the figure and table labels to use the correct nomenclature: Appendix Figure S1-S15 and Appendix Table S1. The term "Supplementary" should not be used. Please ensure that all corresponding callouts in the text are updated accordingly.
- The Table of Contents in the appendix should include page numbers for all listed items.

14. Please do not upload the main figures (Figures 1-7) in .pptx in your next submission.

15. Rename "Materials and Methods" to "Methods". "Main text" heading should be removed.

16. Data availability

- the section heading should be renamed to "Data availability".
- Since the study does not generate large-scale datasets, please only include the following sentence in this section- "This study includes no data deposited in external repositories".

17. The manuscript sections should be in the following order: Title page - Abstract & Keywords - Introduction - Results - Discussion - Methods - Data Availability - Acknowledgments - Disclosure Statement & Competing Interests - References - Figure Legends - (Main Tables with legends if applicable) - Expanded View Figure Legends.

18. Please address the following issues related to figure legends:

- Please note that the figure 4-6 is mislabeled as figure 3-5 in the manuscript. This needs to be rectified.
- Please define the annotated p values ****/**/*/* as well as provide the exact p-values for the same in the legend of figure 5F as appropriate.
- Please note that the exact p values are not provided in the legends of figures 4D, E, H
- Please indicate the statistical test used for data analysis in the legends of figures 5D, E
- Please note that the box plots need to be defined in terms of minima, maxima, centre, bounds of box and whiskers, and percentile in the legends of figures 1B, C, D, E, F, G; 4F-H.
- Please note that information related to n is missing in the legends of figures 1B, C, D, E, F, G; 4E, G, H

Click on the link below to submit your revised paper.

Sincerely,
Jingyi

Jingyi Hou, PhD
Senior Editor
Molecular Systems Biology

If you do choose to resubmit, please click on the link below to submit the revision online before 13th Sep 2025.

***** PLEASE NOTE ***** As part of the EMBO Press transparent editorial process initiative (see our Editorial at <https://dx.doi.org/10.1038/msb.2010.72> , Molecular Systems Biology will publish online a Review Process File to accompany accepted manuscripts. When preparing your letter of response, please be aware that in the event of acceptance, your cover letter/point-by-point document will be included as part of this File, which will be available to the scientific community. More information about this initiative is available in our Instructions to Authors. If you have any questions about this initiative, please contact the editorial office (msb@embo.org).

Reviewer #1:

The authors have addressed my concerns and the manuscript is ready for publication.

Reviewer #3:

In the revised manuscript, the authors include additional published dataset in their analyses of synovial cell states, including healthy synovia from a recent study by Faust et al. However, the manuscript, as presented, does not sufficiently differentiate between original data and published data. I have no problem with reanalysis of published data. Still, as presented, the majority of the figures are not original data, but rather projecting published data in a slightly different manner (i.e. different clustering methods). As such, I do not think this is appropriate for publication as an original research article.

Editor comments:

As you will see below, Reviewer #1 is satisfied with the changes made. However, Reviewer #3 has raised a new concern regarding the originality of the data. Specifically, they note that a substantial portion of the analysis relies on previously published datasets and employs analytical approaches that are not sufficiently differentiated from those used in the original publication.

While this issue was not brought up during the initial round of review, we agree that the current version of the manuscript would benefit from clearer differentiation between the data analysis approach in this study and that of the original publication. In addition, the novel biological findings of your study should be more explicitly emphasized and clearly contextualized within the existing literature. Please address these issues in writing in a revised version of the manuscript.

Many thanks to the editors and reviewers for your comments on the revised version of our paper. We have added new text to the manuscript in the abstract and on pages 7-10 to more clearly differentiate the data analysis approach in this study and that of the original publications, and we have added new text on pages 19-20 to further highlight the novel biological findings of our study in the context of the existing literature. These are all highlighted in the new submitted manuscript file. We have also addressed the editorial points and these too are tracked in the new manuscript and associated appendix files.

Reviewer #1:

The authors have addressed my concerns and the manuscript is ready for publication.

Reviewer #3:

In the revised manuscript, the authors include additional published dataset in their analyses of synovial cell states, including healthy synovia from a recent study by Faust et al. However, the manuscript, as presented, does not sufficiently differentiate between original data and published data. I have no problem with reanalysis of published data. Still, as presented, the majority of the figures are not original data, but rather projecting published data in a slightly different manner (i.e. different clustering methods). As such, I do not think this is appropriate for publication as an original research article.

2nd Sep 2025

Manuscript number: MSB-2025-12951RR

Title: Synovial matrix turnover controls immune cell spatial patterning in inflammation resolution.

Dear Dr. Midwood,

Thank you again for sending us your revised manuscript. We are now satisfied with the modifications made and I am pleased to inform you that your paper has been accepted for publication.

Sincerely,
Jingyi

Jingyi Hou, PhD
Senior Editor
Molecular Systems Biology
